# From Many Imperfect to One Trusted: Imitation Learning from Heterogeneous Demonstrators with Unknown Expertise

## Abstract

Imitation learning (IL) typically depends on large-scale demonstrations collected from *multiple* human or algorithmic demonstrators. Yet, most existing methods assume these demonstrators are either homogeneous or near-optimal—a convenient but unrealistic assumption in many real-world settings. In this work, we tackle a more practical and challenging setting: IL from *heterogeneous* demonstrators with *unknown* and *widely varying expertise levels*. Instead of assuming expert dominance, we model each demonstrator's behavior as a *flexible* mixture of optimal and suboptimal policies, and propose a novel IL framework that jointly learns (a) a state-action optimality scoring model and (b) the latent expertise level of each demonstrator, using only a handful of human queries. The learned scoring model is then integrated into an policy optimization procedure, where it is fine-tuned with offline demonstrations, on-policy rollouts, and a fine-grained mixup regularizer to produce informative rewards. The agent is trained to maximize these learned rewards in an iterative fashion. Experiments on continuous-control benchmarks show that our approach consistently outperforms baseline methods. Even when all demonstrators are *highly suboptimal*, each exhibiting only $5-15\%$ optimality, our method achieves performance comparable to a baseline trained on purely optimal demonstrations, despite our lack of optimality labels.

## 1 Introduction

Imitation learning (IL) offers a convenient framework that enables agents to acquire skills directly from expert demonstrations (Zare et al., 2024). Prior research has shown that effective policies can be learned efficiently when provided with large-scale and high-quality demonstrations (Belkhale et al., 2023; Saxena et al., 2025). However, in real-world scenarios, large-scale datasets are often collected from *multiple* demonstrators, and human or algorithmic demonstrators typically possess strengths in specific contexts and may perform suboptimally outside their areas of expertise. For example, in autonomous driving data collected from electric vehicles (Lee et al., 2022; Zhang et al., 2024), drivers demonstrate a wide range of skill levels, resulting in *heterogene*ous and *imperfect* trajectories with unknown degrees of expertise. Treating all such demonstrators as perfect experts can lead to unreliable or even unsafe agent behavior, posing serious risks in safety-critical applications.

In this work, we study IL from heterogeneous demonstrators whose expertise levels are unknown and variable, naturally leading to datasets containing suboptimal demonstrations. Existing methods (see details in Section 2) addressing this setting often rely on one or more of the following assumptions: (i) demonstrators are homogeneous, i.e., with same expertise levels, and the optimal data dominates the dataset, (ii) access to explicit labels indicating the optimality of demonstrations, or (iii) the expertise level of each demonstrator is known a priori. These requirements are demanding and often impractical when demonstrations come from uncontrolled or large-scale sources.

This brings us to a central question:

*Can we design effective IL algorithms without relying on any of these assumptions?*

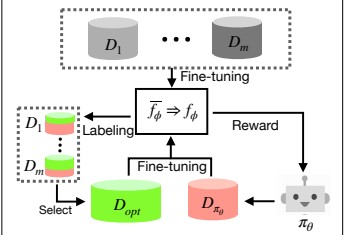

Figure 1: **Overview of our heterogeneous IL framework**, illustrated with $m$ demonstrators possessing (unknown) heterogeneous expertise levels (represented in grayscale; darker shades indicate higher expertise). Green/red indicate inferred optimal/suboptimal pairs. **Stage 1** (left panel): Joint learning of *demonstrator expertise levels* $\{\alpha_i\}_{i=1}^m$ and the *optimality scoring model* $f_\phi$ through a a closed-loop process: (1) a surrogate classifier $\overline{f}_\phi$ is trained to predict the demonstrator identity for each $(s, a)$ pair, which induces $f_\phi$; (2) $\{\hat{\alpha}_i\}_{i=1}^m$ are updated based on predictions from $f_\phi$. **Stage 2** (right panel): Iterative refinement of both $f_\phi$ and the agent policy $\pi_\theta$ using the demonstrator dataset $D = \{(i, D_i)\}_{i=1}^m$, a selected set of optimal demonstrations $D_{\text{opt}}$, and agent-generated samples $D_{\pi_\theta}$.

This problem is inherently challenging because neither demonstrator expertise levels nor demonstration optimality is observed. In the worst case, all demonstrators may exhibit very low expertise, making it extremely difficult to extract reliable signals for policy learning.

Our first contribution is a novel problem formulation for *IL from heterogeneous demonstrators*, where each demonstrator's policy is modeled as a mixture of optimal and suboptimal behaviors, and their *expertise level* is characterized as the proportion of their actions that align with the optimal policy. These expertise levels *vary* across demonstrators, capturing their heterogeneity, but are latent and unknown, leaving little to no direct information about the quality of the demonstrations. Our goal is to learn a high-quality policy from this heterogeneous and suboptimal dataset.

Our second contribution is a novel two-stage methodology for addressing the above problem (Figure 1). In the first stage, we jointly estimate demonstrator expertise levels and learn a *scoring model* that assigns *optimality scores* to the state-action pairs, representing the likelihood that the pair corresponds to the (latent) optimal policy. This is achieved through an EM-style iterative procedure, and we also provide a theoretical analysis showing that this process converges under mild assumptions. In the second stage, the learned scoring model serves as a *surrogate reward function* for policy learning. However, since it is trained entirely offline, it may suffer from covariate shift during policy rollouts. To mitigate this, we introduce a refinement procedure that alternates between policy learning and scoring model updates. This feedback loop produces a more informative reward signal, which is then used to further improve the agent policy.

We evaluate our method in two challenging heterogeneous scenarios: (i) a *wide-range setting*, with demonstrator expertise levels spanning $[0.1, 0.9]$, and (ii) a *low-quality setting*, where all demonstrators are highly suboptimal with expertise in $[0.05, 0.15]$. Across both regimes, our method outperforms baselines and, remarkably, achieves performance comparable to methods trained on the subset of purely optimal demonstrations. These results highlight the practical value of our framework in realistic IL settings where demonstrator quality is unknown, uncurated, and highly variable.

## 2 RELATED WORK

**IL with Homogeneous Demonstrators**    A large body of IL research assumes that demonstrations originate from a single, near-optimal expert policy. This assumption underpins classical approaches such as *behavior cloning* (BC) (Bain & Sammut, 1995), as well as more recent methods like *generative adversarial imitation learning* (GAIL) (Ho & Ermon, 2016), which matches the state–action occupancy measures of the learner and the demonstrator via adversarial training. Even methods designed for noisy demonstrations, such as *robust imitation learning* (RIL) (Tangkaratt et al., 2020), retain this homogeneity assumption by modeling demonstrators' behaviors as generated from a fixed mixture of optimal and suboptimal policies with a constant mixing coefficient representing expertise level, and assume expert data dominance. While effective in controlled benchmarks, these methods may fail in more realistic multi-demonstrator scenarios where expertise levels vary widely.

Table 1: Comparison of IL methods. ✓ = satisfies criterion, ✗ = does not. "Non-dominant opt. demo" means the method does not require optimal demonstrations to dominate the dataset. "Multi-demo support" refers to the ability to handle demonstrations from multiple demonstrators with varying expertise levels. "Handles structured subopt." indicates whether the method can manage structured (i.e., non-random) suboptimal demonstrations. "No opt. labels needed" indicates no labels about the optimality of demonstrations is needed. "Prob. demo scoring" denotes whether the method infers a probabilistic measure of demonstration quality.

| Method | BC | GAIL | RIL | IC-GAIL | PU-GAIL | RCE | WGAIL | ILEED | Ours |
|---|---|---|---|---|---|---|---|---|---|
| Non-dominant opt. demo | ✗ | ✗ | ✗ | ✓ | ✓ | ✗ | ✗ | ✓ | ✓ |
| Multi-demo support | ✗ | ✗ | ✗ | ✗ | ✗ | ✗ | ✓ | ✓ | ✓ |
| Handles structured subopt. | ✗ | ✗ | ✓ | ✓ | ✓ | ✗ | ✓ | ✗ | ✓ |
| No opt. labels needed | ✗ | ✗ | ✓ | ✗ | ✓ | ✗ | ✓ | ✓ | ✓ |
| Prob. demo scoring | ✗ | ✓ | ✓ | ✗ | ✓ | ✓ | ✗ | ✗ | ✓ |

**IL with Optimality Labels**    To better handle suboptimal demonstrations, many IL methods use supervision to distinguish optimal from non-optimal behavior. This strategy underlies extensions of GAIL. For example, *imperfect demonstration and confidence GAIL* (IC-GAIL) (Wu et al., 2019) uses confidence labels to reweight imperfect demonstrations, and *positive-unlabeled GAIL* (PU-GAIL) (Wang et al., 2023) leverages positively labeled success trajectories. Similarly, example-based methods like *recursive classification of examples* (RCE) (Eysenbach et al., 2021) train success classifiers from labeled examples to create proxy rewards. These methods work well when labeled data is available but are unsuitable when such supervision is hard to obtain.

**IL without Homogeneity or Labels**    More recent efforts aim to learn from heterogeneous, unlabeled, and suboptimal demonstrations without relying on any of the above assumptions. *Weighted GAIL* (WGAIL) (Wang et al., 2021) proposes to estimate importance weights directly from discriminator outputs, enabling the policy to emphasize near-optimal samples without requiring prior knowledge of demonstrator quality. *Imitation learning by estimating expertise of demonstrators* (ILEED) (Beliaev et al., 2022) addresses the same challenge by jointly learning a policy and state-dependent expertise levels, treating expertise as a latent variable conditioned on state and demonstrator embeddings and optimized via maximum likelihood. However, their formulation assumes that suboptimal demonstrations arise from a uniformly random distribution, which limits its ability to handle structured suboptimality. Building on this direction, our work defines expertise level as the prior probability that a state–action pair comes from an optimal policy. This formulation accommodates structured suboptimality and eases estimation by avoiding strong state dependence.

## 3  IMITATION LEARNING FROM HETEROGENEOUS DEMONSTRATORS

### 3.1  PROBLEM SETUP

In reinforcement learning (RL), the optimal policy maximizes the expected cumulative reward: $\pi^* = \mathrm{argmax}_\pi \, \mathbb{E}_{p_\pi} \left[ \sum_{t=1}^{T} \gamma^t r(s_t, a_t) \right]$, where $r(s_t, a_t)$ is the reward associated with the state-action pair $(s, a)$ at time $t$, $p_\pi$ is the trajectory distribution induced by policy $\pi$, and $\gamma$ is the discount factor. Unlike RL, in IL the reward function is unobserved and learning relies on a dataset of demonstrator trajectories $\tau = \{(s_0, a_0), ..., (s_{T_\tau}, a_{T_\tau})\}$ of varying lengths $T_\tau$. The goal is to learn a policy $\pi$ whose induced state-action distribution matches that of the optimal policy (Ziebart et al., 2010).

To make this feasible, most IL algorithms assume that *all* demonstrations are generated by the optimal policy. However, this assumption is overly restrictive, as real datasets are often collected from multiple demonstrators with varying skills. In this work, we study a more practical IL setting where demonstrations are provided by heterogeneous demonstrators with different expertise levels.

**Dataset:** We consider $m$ demonstrators. Each demonstrator $i \in [m]$ provides a set of trajectories $D_i$ and is associated with an *expertise level* $\alpha_i \in [0, 1]$. We define $\alpha_i$ as the probability that any given $(s, a)$ pair from $D_i$ was generated by the optimal policy, i.e., $\alpha_i = \mathbb{P}[(s, a) \sim \pi^* | (s, a) \in D_i]$. This represents a demonstrator-level attribute: a higher $\alpha_i$ means that demonstrator $i$ tends to act more consistently with $\pi^*$. We consider *demonstrator heterogeneity*, i.e., there $\exists \, i, j \in [m], i \neq j$, such that $\alpha_i \neq \alpha_j$. The full dataset is $D = \{(i, D_i)\}_{i=1}^{m}$, and the expertise levels $\{\alpha_i\}_{i=1}^{m}$ are *unknown*.

**Optimality Score:** For each $(s, a)$ pair, we introduce a binary latent variable $z \in \{0, 1\}$ indicating its *optimality*: $z = 1 \Rightarrow (s, a) \sim \pi^*$, while $z = 0 \Rightarrow (s, a) \sim \pi^{\text{sub}}$, where $\pi^{\text{sub}}$ denotes an unknown suboptimal policy. The *optimality score* is defined as the probability that $(s, a)$ is optimal given its values: $\mathbb{P}(z = 1 | s, a)$, which can be seen as a sample-specific, conditional expertise level.

Let $\rho^{\pi^*}(s)$ and $\rho^{\pi^{\text{sub}}}(s)$ denote the state visitation distribution under $\pi^*$ and $\pi^{\text{sub}}$, respectively. Since demonstrators are heterogeneous, for each demonstrator-$i$, we model their demonstrations using the statistical mixture[1]

$$\rho_i(s, a) = \alpha_i \underbrace{\rho^{\pi^*}(s)\, \pi^*(a \mid s)}_{P(s,a|z=1):=\rho^{\pi^*}(s,a)} + (1 - \alpha_i) \underbrace{\rho^{\pi^{\text{sub}}}(s)\pi^{\text{sub}}(a \mid s)}_{P(s,a|z=0):=\rho^{\pi^{\text{sub}}}(s,a)}. \tag{1}$$

Here, $P(s, a \mid z = 1)$ and $P(s, a \mid z = 0)$ are the occupancy measures of the optimal and suboptimal policies. This formulation corresponds to each demonstrator generating samples by first drawing $z \sim \text{Bernoulli}(\alpha_i)$ and then sampling $(s, a) \sim P(s, a \mid z)$. It ensures that all demonstrators share the same optimal and suboptimal occupancy components, differing only in expertise level $\alpha_i$.

**Policy Learning:** Given demonstrations $D$ from these heterogeneous demonstrators, our goal is to match the optimal policy $\pi^*$. Without access to rewards, we aim to learn a *scoring model* $f_\phi$ : $\mathcal{S} \times \mathcal{A} \to \mathbb{R}$, where $\mathcal{S}$ denotes the set of states and $\mathcal{A}$ denotes the set of actions, parameterized by $\phi$ to estimate the optimality scores $\mathbb{P}(z = 1 | s, a)$ and then use it as a surrogate reward in RL: $\hat{\pi}^* = \arg\max_\pi \mathbb{E}_{p_\pi} \left[ \sum_{t=1}^T \gamma^t f_\phi(s_t, a_t) \right]$.

### 3.2 JOINT ESTIMATION OF EXPERTISE LEVELS AND SCORING MODEL

We now describe how to learn the scoring model $f_\phi$ directly from the heterogeneous dataset $D$.

#### 3.2.1 SURROGATE DEMONSTRATOR CLASSIFICATION FOR OPTIMALITY SCORING

The optimality scoring task can be framed as a binary *class probability estimation* (Buja et al., 2005) problem, where we aim to estimate $\mathbb{P}(z = 1 | s, a)$ for each $(s, a)$ pair being optimal (with label $z = 1$) or sub-optimal (with label $z = 0$). In the supervised setting, if explicit optimality labels $z$ were available, one could directly train $f_\phi$ with a proper scoring rule, such as cross-entropy. However, in our setting, such labels are not available. To solve this problem, we adopt a surrogate learning approach inspired by the surrogage set classification (SSC) framework of Lu et al. (2021).

**Surrogate Demonstrator Classification** We use the demonstrator identity $\bar{z} \in [m]$ as a surrogate label. Then the surrogate task involves training a multi-class classifier $\overline{f}_\phi : \mathcal{S} \times \mathcal{A} \to \mathbb{R}^m$ to predict the demonstrator from which each $(s, a)$ pair originated, by minimizing the risk:

$$\mathcal{L}(\overline{f}_\phi) = \mathbb{E}_{(s,a) \sim \sum_i \mu_i \rho_i(s,a)}[\ell(\overline{f}_\phi(s, a), \bar{z})], \tag{2}$$

where $\mu_i$ is the prior probability of sampling from demonstrator $i$, estimated as $\mu_i \approx \frac{||D_i||}{\sum_{i=1}^m ||D_i||}$, $||D_i||$ denotes the cardinality of the dataset $D_i$, and $\ell$ is the cross-entropy loss. For simplicity, we denote $\rho := \sum_i \mu_i \rho_i(s, a)$.

**Recovering Optimality Scores** Under demonstrator heterogeneity, Lu et al. (2021) show that the optimality scoring model $f_\phi$ can be recovered from the surrogate classifier $\overline{f}_\phi$ via an injective transformation $T(\cdot) : \mathbb{R} \to \mathbb{R}^m$, such that: $\overline{f}_\phi = T_i(f_\phi)$,[2] where the $i$-th component of $T$ is

$$T_i(x) = \frac{\mu_i(\alpha_i - \alpha')x + \mu_i\alpha'(1 - \alpha_i)}{\sum_{i=1}^m \mu_i(\alpha_i - \alpha')x + \sum_{i=1}^m \mu_i\alpha'(1 - \alpha_i)}, \tag{3}$$

---

[1] We note that $\rho_i(s, a)$ is a *statistical* mixture over demonstrator-generated samples, rather than the occupancy induced by executing $\pi_i$ in the MDP. In particular, its marginal $\rho_i(s) = \sum_a \rho_i(s, a) = \alpha_i \rho^{\pi^*}(s) + (1 - \alpha_i)\rho^{\pi^{\text{sub}}}(s)$ generally differs across demonstrators and is *not* equal to the MDP visitation distribution $\rho^{\pi_i}(s)$. This mixture model is used only to describe demonstration generation process, not to define the learner's policy.

[2] With a slight abuse of notation, we use $f_\phi$ and $\overline{f}_\phi$ to refer to both classifiers and class probability estimators, i.e., scoring functions in the sense of Buja et al. (2005).

---

**Algorithm 1** EM-style Joint Learning of Expertise Levels and Scoring Model

---

1: **Input**: Dataset $D = \{(i, D_i)\}_{i=1}^m$, number of sample queries $n$
2: **repeat**
3:     **Initialization**: Scoring model $f_\phi(s, a)$, expertise levels $\{\alpha_i\}_{i=1}^m \sim \mathcal{U}(0.1, 0.9)$, $\forall i \in [m]$
4:     **for** $t = 1, 2, ..., T$ **do**
5:         **M-step (Minimize risk to update scoring model):**
6:         Update $f_\phi$ by minimizing the empirical risk from Eq. 4 using current $\{\alpha_i\}_{i=1}^m$
7:         **E-step (Estimate expertise levels):**
8:         Update expertise levels $\alpha_i \leftarrow \frac{\sum_{(s,a) \in D_i} [f_\phi(s,a) > 0.5]}{||D_i||}$, $\forall i \in [m]$
9:     **end for**
10: **until Expertise Dispersion Criterion** is satisfied
11: Query $n$ samples and correct predictions to satisfy **Optimality Alignment Criterion**
12: **Return** $\{\alpha_i\}_{i=1}^m$, $f_\phi$

---

Here, $\alpha_i = \mathbb{P}(z = 1 | \bar{z} = i)$ is the expertise level of demonstrator $i$, and $\alpha'$ is the expected expertise in the target environment where the scoring model is deployed.[3] If these expertise levels are known, $f_\phi$ can be learned indirectly by training $\bar{f}_\phi$ via Eq. 2, which induces updates on $f_\phi$ through the following equivalent objective

$$\mathcal{L}(\phi) = \mathbb{E}_{(s,a) \sim \rho} \left[ \ell(T(f_\phi(s, a)), \bar{z}) \right]. \tag{4}$$

**Lemma 1** (Optimal Scoring Model Recovery, Lu et al. (2021)). *Assume the model class is well-specified, $\ell$ is a proper loss (e.g., cross-entropy), and $T$ is correctly computed as in (3). Let $\bar{f}_\phi$ be the surrogate classifier trained optimally according to (4). Then the induced binary scoring model $f_\phi$ accurately estimates the optimality scores: $f_\phi(s, a) = \mathbb{P}(z = 1 \mid s, a)$.*

With the optimal scoring model, we can guarantee that the RL policy $\hat{\pi}^*$ learned using the surrogate reward $f_\phi$ matches the optimal policy $\pi^*$ in both discounted occupancy and action distribution.

**Theorem 2** (Optimal Policy Recovery). *Let $\pi^*$ be the unknown optimal policy. Define the discounted occupancy of any policy $\pi$ by $\rho^\pi(s, a) = \mathbb{E}_{\tau \sim p_\pi} \left[ \sum_{t=1}^T \gamma^t \mathbf{1}_{\{(s_t, a_t) = (s, a)\}} \right]$, where $\mathbf{1}_{\{\cdot\}}$ is the indicator function. Let the surrogate RL objective be*

$$J_f(\pi) = \mathbb{E}_{p_\pi} \left[ \sum_{t=1}^T \gamma^t f_\phi(s_t, a_t) \right] = \sum_{s,a} \rho^\pi(s, a) f_\phi(s, a).$$

*Assume the scoring model is optimal, i.e., $f_\phi(s, a) = \mathbb{P}(z = 1 \mid s, a)$ for all $(s, a)$. Then any maximizer $\hat{\pi}^* = \arg\max_\pi J_f(\pi)$ satisfies $\rho^{\hat{\pi}^*}(s, a) = \rho^{\pi^*}(s, a)$ for all $(s, a)$ in the support of $\rho^{\pi^*}$. If, in addition, the demonstration distribution satisfies the support coverage condition $\mathrm{supp}(\rho^{\pi^*}) \subseteq \mathrm{supp}(\rho)$, then for every state $s$ with $\rho^{\pi^*}(s) > 0$, the learned policy recovers the optimal policy exactly: $\hat{\pi}^*(a \mid s) = \pi^*(a \mid s)$ for all $a$ in the support of $\rho^{\pi^*}(s, \cdot)$.*

### 3.2.2 EM-STYLE OPTIMIZATION OF SCORING MODEL AND EXPERTISE ESTIMATES

However, neither optimality labels nor expertise levels are observed in our setting. To jointly learn both, we propose an expectation-maximization (EM)-style algorithm, outlined in Algorithm 1.[4]

The algorithm begins by randomly initializing the expertise levels $\{\alpha_i\}_{i=1}^m$ within the range $[0.1, 0.9]$. Given the initial $\{\alpha_i\}_{i=1}^m$, we update the scoring model $f_\phi$ using the surrogate objective Eq. 4. We then refine the expertise levels by computing the fraction of samples from each demonstrator that are labeled as optimal by the current scoring model. This alternating update procedure is repeated for $T$ epochs. Here, we present a convergence result that guarantees the EM-style optimization of of $\alpha$ and $\phi$ converges to a stationary point. The proof is in Appendix D.

---

[3]Since this expected expertise is unknown, we set $\alpha'$ as 0.5 in this work. Our ablation studies in Tab. 15-16, and Fig. 7 justify this choice.

[4]To make the expertise level estimation problem solvable, we assume the existence of anchor points in the demonstration dataset, i.e., data points that belong to class $z = 1$ and $z = 0$ almost surely.

---

**Algorithm 2** Joint Learning of Scoring Model and Agent Policy with Demonstration Relabeling

---

1: **Input**: Dataset $D = \{(i, D_i)\}_{i=1}^m$, learned scoring model $f_\phi(s, a)$, estimated expertise levels $\{\hat{\alpha}_i\}_{i=1}^m$, ratio of pseudo-optimal samples $k$
2: **Initialization**: early_stop ← False, agent policy $\pi_\theta$, $D_{\text{opt}} \leftarrow \{(s, a) \mid f_\phi(s, a) > 0.5\}$
3: **for** $t = 1, 2, ..., T$ **do**
4:     Sample trajectories from agent: $D_{\pi_\theta} \leftarrow \{(s, a) \mid (s, a) \sim \pi_\theta\}$
5:     Update $f_\phi$ by minimizing the empirical risk from Eq. 6
6:     **if** not early_stop **then**
7:         Relabel pseudo-optimal set: $D_{\text{opt}} \leftarrow \text{top-}k(s, a) \in D$ by $f_\phi(s, a)$
8:         Check **early stopping criterion** and update early_stop if necessary
9:     **end if**
10:    Update $\pi_\theta$ using RL with reward $\log(f_\phi(s, a))$
11: **end for**
12: **Return** final policy $\pi_\theta$

---

**Theorem 3** (Convergence Guarantee). *Consider the optimization of the objective $\mathcal{L}(\phi, \alpha)$ via iterative updates of the scoring model $f_\phi$ and the expertise levels $\alpha = \{\alpha_i\}_{i=1}^m$. Let $(\phi^t, e^t)$ be the parameters at iteration $t$. If the updates follow:*

$$\phi^{t+1} = \arg\min_\phi \mathcal{L}(\phi^t, \alpha^{t+1}), \quad \alpha^{t+1} = \frac{\sum_{(s,a) \in D_i} [f_{\phi^t}(s, a) > 0.5]}{||D_i||} \tag{5}$$

*where $\mathcal{L}(\phi, \alpha) = \mathbb{E}_{(s,a)\sim\rho}[\ell(T_\alpha(f_\phi(s, a)), \bar{z})]$, then the sequence $\{\mathcal{L}(\phi^t, \alpha^t)\}_{t=1}^\infty$ is monotonically non-increasing, i.e., $\mathcal{L}(\phi^{t+1}, \alpha^{t+1}) \leq \mathcal{L}(\phi^t, \alpha^t)$, ensuring convergence to a stationary point.*

While Theorem 3 guarantees convergence, the quality of the resulting solution is sensitive to the initialization of $\{\alpha_i\}_{i=1}^m$. As with other EM-style procedures, it may converge to suboptimal solutions depending on the initialization. To mitigate this, we run the algorithm with multiple random initializations until Expertise Dispersion Criterion is satisfied.

**Expertise Dispersion Criterion** Since our algorithm relies on self-labeled data during training, it is susceptible to error propagation and confirmation bias. Empirically, we observe that certain random initializations lead to degenerate solutions, where the scoring model becomes overly confident and the estimated expertise levels collapse to nearly identical values. To assess the reliability of the learned expertise levels $\{\alpha_i\}_{i=1}^m$, we measure their *dispersion* by computing the variance $\text{Var}(\{\alpha_i\}_{i=1}^m)$. If the variance falls below a small threshold $\epsilon > 0$, we consider the model to have failed in distinguishing between different expertise levels, as the predictions carry little information, analogous to having low entropy, and are indicative of overfitting. Conversely, a higher variance suggests more informative and diverse estimates, suggesting the model has captured demonstrator heterogeneity. We therefore use this variance as a selection criterion to retain only solutions with sufficient dispersion in $\{\alpha_i\}_{i=1}^m$.

**Optimality Alignment Criterion** Due to the unsupervised nature of our setting, the learned scoring model may inadvertently invert the notion of optimality. In such cases, the estimated expertise levels $\alpha_i$ may effectively correspond to $1 - \alpha_i$. To detect and correct this issue, we query a small subset of state–action pairs (in our experiments, as few as 5 samples suffice) and ask an expert to verify the correctness of their predicted labels. If the majority of results indicate a systematic inversion in predicted labels, we simply flip the predicted labels and correct the outputs as follows:

$$\alpha_i \leftarrow 1 - \alpha_i, \ f_\phi(s, a) \leftarrow 1 - f_\phi(s, a), \ \forall i \in [m], \ \forall (s, a) \in D.$$

### 3.3 IMITATION LEARNING VIA SCORING MODEL

After learning the scoring model $f_\phi$ and estimating the expertise levels $\{\alpha\}_{i=1}^m$, we perform IL.

#### 3.3.1 PROGRESSIVE SCORING MODEL REFINEMENT VIA FINE-GRAINED SAMPLE SELECTION

The scoring model $f_\phi$ is initially trained only on demonstration data, which limits its generalization to learner-generated trajectories due to covariate shift (Chang et al., 2021). To improve its robustness, we refine $f_\phi$ during agent learning using three complementary terms: *demonstration discriminative loss* in Eq. 4, reflecting the original training signal; *agent-matching penalty*, discouraging the

misclassification of learner-generated behaviors as optimal; and a *mixup regularizer* (Zhang et al., 2017) which interpolates between agent and demonstrator behaviors to smooth the scoring boundary and stabilize training. The overall objective for refining $f_\phi$ is given by:

$$\min_{f_\phi} \widehat{\mathbb{E}}_{(s,a)\sim\rho} \left[ \mathcal{L}(f_\phi(s,a) \right] + \widehat{\mathbb{E}}_{(s',a')\sim\rho^{\pi_\theta}} \left[ \log(f_\phi(s',a')) \right] + \widehat{\mathbb{E}}_{\substack{(s,a)\sim\hat{\rho}^{\pi^*} \\ (s',a')\sim\rho^{\pi_\theta}}} \left[ \mathcal{L}_{\text{mixup}}(s,a,s',a') \right] \quad (6)$$

where $\widehat{\mathbb{E}}$ denotes empirical expectation; $\hat{\rho}^{\pi^*}$ represents pseudo-optimal data inferred from $D_{\text{opt}} \subseteq D$;

$$\mathcal{L}_{\text{mixup}}(s,a,s',a') = \lambda \log \left( 1 - f_\phi(\tilde{s},\tilde{a}) \right) + (1-\lambda) \log \left( f_\phi(\tilde{s},\tilde{a}) \right),$$
$$\tilde{s} = \lambda s + (1-\lambda)s', \ \tilde{a} = \lambda a + (1-\lambda)a', \ \lambda \sim \mathcal{U}(0,1).$$

Early in training, the novice agent tends to produce suboptimal behaviors. The agent-matching penalty discourages $f_\phi$ from assigning high scores to these early samples, thereby enhancing its ability to distinguish between agent and demonstrator behavior and reducing covariate shift. In parallel, the mixup regularizer acts as both data augmentation and regularization based on agent and expert data, yielding more stable and informative gradients, ultimately improving the effectiveness of policy learning in IL (Orsini et al., 2021). The full training procedure is outlined in Algorithm 2.

To further refine $f_\phi$, we adopt a bootstrapping approach where $f_\phi$ is used to pseudo-label demonstration data $D$. At each iteration: (1) we extract a pseudo-optimal subset $D_{\text{opt}} \subseteq D$ by applying a *top-k selection method*, where we choose the lowest $\lfloor k \cdot \|D\| \rfloor$ ($\lfloor \cdot \rfloor$ denotes the floor function) state–action pairs according to their $f_\phi$ scores; (2) we generate new agent trajectories $D_{\pi_\theta}$ to represent $\rho^{\pi_\theta}$ ; (3) we update $f_\phi$ using the loss in Eq. 6 (Line 5), computed over $D$, $D_{\pi_\theta}$, and $D_{\text{opt}}$; (4) we relabel $D_{\text{opt}}$ using the updated $f_\phi$ (Line 7). This mutual reinforcement between scoring model refinement and pseudo-label improvement allows both components to jointly bootstrap toward higher fidelity.

### 3.3.2 POLICY LEARNING WITH SCORING MODEL AS A SURROGATE REWARD

As training progresses, the agent begins to generate increasingly competent behaviors. However, these samples are still treated as suboptimal during scoring model refinement. Notably, at this stage, the scoring model $f_\phi$ functions similarly to a discriminator in the GAIL framework (Ho & Ermon, 2016), struggling to distinguish between optimal and agent-generated data. Despite this, our approach is fundamentally different from GAIL: while GAIL relies exclusively on optimal demonstrations for supervision, our method operates in an (almost) unsupervised setting, allowing $f_\phi$ to learn from a large number of suboptimal demonstrations provided by heterogeneous demonstrators. This makes our approach particularly advantageous in scenarios where demonstration quality is low.

As the agent improves, $f_\phi$ becomes increasingly confused, and its predictions become unreliable. To prevent the accumulation of incorrect labels that could mislead policy learning, we introduce an **early stopping criterion** for the relabeling process: let $\Delta_t$ denote the number of sample changes in $D_{\text{opt}}$ between iterations $t$ and $t-1$, counting both removed and newly added samples; if $\Delta_t < \delta$ for $p$ consecutive steps, we stop relabeling and freeze $D_{\text{opt}}$. The updated $f_\phi$ is employed as a surrogate reward function for training the agent policy $\pi_\theta$ (Line 10). Specifically, we use $\log f_\phi(s,a)$ as the reward signal in reinforcement learning. This policy is updated using soft actor-critic (SAC) (Haarnoja et al., 2018) iteratively, alternating with scoring model refinement and pseudo-label updates.

## 4 EXPERIMENTS

We will test whether our algorithm can (1) train a scoring model on multiple sets of imperfect demonstration data collected from heterogeneous demonstrators in order to provide optimality labels and estimate demonstrator expertise levels, and (2) learn an optimal agent policy based on the scoring model and imperfect demonstrations.

To evaluate the robustness of our algorithm under widely varying and low demonstrator expertise, we design two experiments: (a) **General expertise test**, using demonstrations from demonstrators with expertise levels in a normal wide range $[0.1-0.9]$ (average optimal ratio 0.5), and (b) **Low expertise test**, with extremely low expertise levels in $[0.05-0.15]$ (average optimal ratio 0.1). Prior works have mostly focused on cases where most demonstrations are optimal, with only limited exploration of low-optimal-ratio settings, where their performance degrades significantly (Eysenbach et al., 2021; Wang et al., 2021; 2023). To address this gap, we specifically evaluate the low expertise setting.

In both tests, datasets are constructed from six demonstrators (each providing 5,000 samples) with uniformly distributed expertise levels (see Tab. 6 in Appendix). We set the top-$k$ hyperparameter $k$ to match the average optimal ratio in each test.

## 4.1 ENVIRONMENT AND DATASETS

We evaluate our algorithm against baseline methods on four MuJoCo environments (Todorov et al., 2012; Towers et al., 2024) and conduct ablation studies to assess the contributions of our proposed components. Demonstrations are collected by training a policy with SAC with the true reward. An early-stage checkpoint $\pi^{\text{sub}}$ provides suboptimal demonstrations, while the final checkpoint $\pi^*$ provides optimal demonstrations (see Tab. 5 for the checkpoints' performance). As in Beliaev et al. (2022); Tangkaratt et al. (2020), the imperfect dataset is mixture of state-action pairs from $\pi^*$ and $\pi^{\text{sub}}$. Unlike random behaviors, our suboptimal data comes from a real suboptimal policy, which reflects that experts may make mistakes, but their actions are still better than random actions. Using this approach, multiple imperfect demonstration sets $D = \{(i, D_i)\}_{i=1}^m$ are collected.

## 4.2 COMPARISON WITH BASELINES

Across the experiments, we benchmark our model against four IL baseline algorithms. (1) **GAIL o.s. (optimal subset)** (Ho & Ermon, 2016): GAIL trained on the purely optimal subset of demonstrations, serving as an oracle method; (2) **GAIL**: GAIL trained on all (optimal and suboptimal) demonstrations; (3) **RIL** (Tangkaratt et al., 2020): trains an optimality discriminator via pseudo-labeling with co-training; (4) WGAIL (Wang et al., 2021): estimates importance weights to emphasize near-optimal demonstrations. (5) ILEED (Beliaev et al., 2022): learns the policy in a offline training manner via maximum likelihood. (6) BC (Bain & Sammut, 1995). For fair comparison, we adopt a consistent model architecture for discriminator/classifier/scoring model, and learner policy, modifying only the input and output layers to match the corresponding state and action spaces of each environment. Moreover, all online baselines incorporate mixup regularization. All learner policies are updated using SAC with same hyperparameters. Training details are provided in Tab. 21 in the Appendix. We use 5 random seeds per condition and report the mean and standard deviation in all our results. For offline methods (ILEED and BC), for each random seed, we repeat the algorithm 10 times with random initializations and report the best result by selecting the run with the maximal likelihood.

The results are summarized in Tab. 2 (see Fig. 3 in the Appendix for training curves). In the general expertise test, our algorithm outperforms all baselines except GAIL o.s., achieving at least twice the rewards in all tasks. While GAIL o.s. attains comparable rewards, our algorithm still yields slightly higher reward in Ant, Swimmer and Walk2d. This is attributed to that our algorithm can achieve high classification performance on imperfect demonstrations (see Tab. 4), in contrast to GAIL o.s., is also capable of leveraging suboptimal data and perform better in the subsequent IL stage. In the low xpertise test, the trends remain similar: our algorithm surpasses all baselines except GAIL o.s. However, decreased classification performance on imperfect demonstrations (see Tab. 4) reduces overall rewards compared to the general expertise test. In this setting, GAIL o.s., which trains on purely optimal subset of demonstrations, achieves higher rewards in most tasks.

The low rewards achieved by other baselines in both experiments can be attributed to the several factors. For GAIL, which treats all demonstrations as optimal, its performance is limited when handling imperfect demonstrations. RIL assumes that most demonstrations are optimal, but in our experiments, at least half are suboptimal. Consequently, the discriminator's classification risk does not align with the risk of distinguishing between optimal and suboptimal demonstrations, reducing the reliability of its predictions for policy learning. Moreover, even with co-training for pseudo-labeling, the overconfidence can only be partially mitigated; once error accumulation begins, it continues to degrade the discriminator's ability to predict optimality of demonstrations. WGAIL requires an optimal discriminator to compute importance weights, which is approximated by assuming that the discriminator in the early stage of GAIL is near-optimal. Although early stopping is used for weight estimation, this assumption can fail due to the instability of adversarial learning.

Table 2: Performance comparison with baselines, mean $\pm$ std of reward across last five checkpoints

| Environment | Dataset | GAIL o.s. | GAIL | RIL | WGAIL | BC | ILEED | Scoring (ours) |
|---|---|---|---|---|---|---|---|---|
| **General expertise test** | | | | | | | | |
| Ant-v4 | $4016.3 \pm 2776.8$ | $5660.4 \pm 163.6$ | $1387.2 \pm 301.0$ | $101.8 \pm 118.0$ | $1744.0 \pm 919.1$ | $1341.8 \pm 164.5$ | $1428.2 \pm 80.3$ | $\mathbf{6084.6 \pm 146.9}$ |
| HalfCheetah-v4 | $9988.9 \pm 5959.9$ | $9278.2 \pm 496.5$ | $3035.8 \pm 1186.4$ | $1631.6 \pm 959.7$ | $4336.2 \pm 335.6$ | $2318.1 \pm 521.2$ | $1526.2 \pm 1020.4$ | $\mathbf{8751.2 \pm 1797.9}$ |
| Swimmer-v4 | $190.7 \pm 149.1$ | $293.6 \pm 28.2$ | $76.6 \pm 28.3$ | $28.8 \pm 16.3$ | $115.6 \pm 83.7$ | $21.3 \pm 23.0$ | $16.8 \pm 20.5$ | $\mathbf{305.4 \pm 15.7}$ |
| Walker2d-v4 | $3096.7 \pm 2549.6$ | $3703.0 \pm 1011.9$ | $948.6 \pm 297.2$ | $1187.2 \pm 130.1$ | $665.4 \pm 330.2$ | $308.4 \pm 111.5$ | $440.3 \pm 313.0$ | $\mathbf{3959.2 \pm 671.4}$ |
| **Low expertise test** | | | | | | | | |
| Ant-v4 | $1816.4 \pm 1726.3$ | $5542.0 \pm 168.1$ | $1439.6 \pm 202.0$ | $2.2 \pm 174.1$ | $971.6 \pm 591.0$ | $1473.4 \pm 97.7$ | $1376.5 \pm 62.1$ | $\mathbf{5932.0 \pm 335.2}$ |
| HalfCheetah-v4 | $5222.0 \pm 3578.8$ | $9194.2 \pm 848.1$ | $3162.8 \pm 1417.2$ | $964.2 \pm 1100.1$ | $3975.6 \pm 369.9$ | $3753.9 \pm 220.5$ | $3767.6 \pm 176.9$ | $\mathbf{7335.0 \pm 1316.4}$ |
| Swimmer-v4 | $71.4 \pm 89.5$ | $244.4 \pm 113.5$ | $37.4 \pm 10.5$ | $44.6 \pm 2.9$ | $48.8 \pm 53.7$ | $27.8 \pm 11.6$ | $34.0 \pm 2.4$ | $\mathbf{195.4 \pm 92.1}$ |
| Walker2d-v4 | $1062.8 \pm 1546.5$ | $3532.6 \pm 782.2$ | $737.8 \pm 163.2$ | $1140.0 \pm 319.0$ | $703.8 \pm 291.2$ | $436.6 \pm 69.1$ | $605.3 \pm 94.5$ | $\mathbf{2961.8 \pm 339.8}$ |

Table 3: Estimation errors of Stage 1 scoring model for expertise levels (mean $\pm$ std).

| Experiment | Ant-v4 | HalfCheetah-v4 | Hopper-v4 | Swimmer-v4 | Walker2d-v4 |
|---|---|---|---|---|---|
| General expertise test | $(2.67 \pm 1.28) \times 10^{-4}$ | $(2.00 \pm 2.00) \times 10^{-5}$ | $(3.83 \pm 2.73) \times 10^{-3}$ | $(2.00 \pm 2.00) \times 10^{-5}$ | $(3.27 \pm 3.12) \times 10^{-4}$ |
| Low expertise test | $(9.65 \pm 0.24) \times 10^{-2}$ | $(3.67 \pm 0.15) \times 10^{-2}$ | $(1.05 \pm 0.02) \times 10^{-1}$ | $(2.81 \pm 0.11) \times 10^{-2}$ | $(1.34 \pm 0.04) \times 10^{-1}$ |

## 4.3 ESTIMATION OF EXPERTISE LEVELS

Tab. 3 reports the estimation errors of the scoring model trained in Stage 1 w.r.t. expertise levels relative to the ground truth, with detailed results provided in Tab. 7-8 in the Appendix. In the general expertise test, where true expertise levels are widely distributed within [0.1-0.9], errors are extremely low, remaining below 0.004 across all tasks. In contrast, in the low expertise test, with levels [0.05-0.15], the estimation errors increase to around 0.1 for most tasks. This indicates that the scoring model tends to be over-optimistic and produce more false-positive labels on imperfect demonstrations (see Tab. 8). However, these incorrect labels are largely corrected during Stage 2.

## 4.4 OPTIMALITY PREDICTION ON IMPERFECTION DEMONSTRATIONS

We evaluate the scoring model $f_\phi$ on its ability to classify imperfect demonstrations in terms of optimality. Labeling is conducted in two stages: Stage 1 using the pretrained $f_\phi$ and Stage 2 using the fine-tuned $f_\phi$. Performance is measured by accuracy and precision, with emphasis on precision, as it reflects the false positive (FP) risk, which is more critical than false negative risk for policy learning (Irpan et al., 2019).

Results in Tab. 4 show that in the general expertise test, both accuracy and precision are nearly 1 in Stage 1, leaving almost no room for improvement in Stage 2. In contrast, in the low expertise test, where expertise levels are narrowly distributed, the accuracy in Stage 1 maintains above 86%, but the precision drops significantly, to 43-78%. This indicates that surrogate demonstrator classification training handles datasets with widely varying expertise well, but struggles when expertise levels are similar. However, Stage 2 improves both accuracy and precision, with precision increasing by 12-115% across tasks. This shows that for imperfect demonstrations with extremely low expertise, many false positives from a poorly pretrained $f_\phi$ can be corrected in Stage 2 after fine-tuning $f_\phi$. Detailed classification results are presented in Tab. 8 in the Appendix.

## 4.5 ABLATION TESTS

To assess the effect of each component of our algorithm, we conduct ablation experiments on the Ant-v4 environment.

**Relabeling and Early Stopping**: We evaluate the effects of the relabeling step (Alg. 2, step 7) and early stopping mechanism (Alg. 2, step 8) in Stage 2, shown in the subfigure (a)-(b) of Fig. 2 (see Tab. 9 for numerical results). Without relabeling, performance is comparable to the original algorithm in the general expertise test but drops by about half in the low expertise test, due to numerous FP errors generated in Stage 1 (see Sec. 4.4). Without the relabeling step in Stage 2, these critical FP errors cannot be corrected using the fine-tuned $f_\phi$, causing the performance drop. Without early stopping during relabeling, rewards decline in both general and low expertise tests as the agent approaches the optimal policy. This occurs because $f_\phi$ can be "fooled" when the agent generates near-optimal samples, increasing classification errors and accumulating incorrect labels, which reinforces overconfidence of $f_\phi$ and corrupts the surrogate reward for the agent.

**Top-$k$ $k$ Value Selection:** We study how different $k$ values could influence the final performance of IL. Results in subfigure (c)-(d) in Fig. 2 show that $k$ can be treated as a hyperparameter without prior knowledge of the distribution of expertise (numerical results in Tab. 10). For both the general

Table 4: Classification results of the scoring model on imperfect demonstrations for Stage 1 and Stage 2 (Accuracy / Precision, %), with Stage 1 → Stage 2 improvement (%) after relabeling.

| Test | Stage | Ant-v4 | Impr. | HalfCheetah-v4 | Impr. | Swimmer-v4 | Impr. | Walker2d-v4 | Impr. |
|------|-------|--------|-------|----------------|-------|------------|-------|-------------|-------|
| General expertise | 1 | 99.91 / 99.89 | - | 100.00 / 100.00 | - | 100.00 / 100.00 | - | 99.92 / 99.90 | - |
| | 2 | 99.89 / 99.89 | -0.02 / 0.00 | 100.00 / 100.00 | 0.00 / 0.00 | 99.94 / 99.94 | -0.06 / -0.06 | 99.94 / 99.94 | 0.02 / 0.04 |
| Low expertise | 1 | 90.33 / **51.78** | - | 96.33 / **73.68** | - | 97.19 / **78.84** | - | 86.39 / **43.01** | - |
| | 2 | 99.65 / **98.23** | 10.32 / **89.71** | 99.98 / **99.91** | 3.79 / **35.60** | 97.58 / **87.91** | 0.40 / **11.50** | 98.48 / **92.40** | 13.99 / **114.83** |

(a) General expertise test    (b) Low expertise test    (c) General expertise test    (d) Low expertise test

Figure 2: Ablation results on the impact of Stage 2 relabeling and early stopping, and $k$ selection

and low expertise tests, where the average optimality ratio (defined as $k^*$) of demonstrations is 0.5 and 0.1 respectively, when $k < k^*$ rewards remain comparable to that when $k = k^*$. However, the performance declines as $k$ increases when $k > k^*$. The reason is that the pseudo optimal set is selected based on the lowest $\lfloor k \cdot ||D|| \rfloor$ state–action pairs by $f_\phi$ scores. Assuming $f_\phi$ is well trained, a $k < k^*$ leads to a conservative selection (subset of the true optimal set), while $k > k^*$ can introduce more FP samples to the pseudo optimal set. This indicates that FP risk is more critical, and a smaller $k$ is preferable.

**Number of Demonstrators:** We conduct an ablation study on the number of demonstrators (each providing 5,000 samples). Results shows that our methods is robust to varying demonstrator counts in terms of IL performance (see Fig. 4 and Tab. 11- 12 in the Appendix for details).

Appendix E.2 compares our method with variants (e.g., oracle settings, PN loss), showing it consistently matches or outperforms them, demonstrating its robustness and effectiveness.

## 5 CONCLUSIONS

We presented a general IL framework for handling unlabeled, imperfect demonstrations from heterogeneous demonstrators with unknown expertise. Our approach combines EM-style training with a refinement step to learn a scoring model that guides policy learning. Experiments across challenging settings demonstrate strong performance, especially in low-quality demonstration regimes, highlighting the value of leveraging suboptimal data in an (almost) unsupervised manner.

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

# Appendix

## A    FURTHER DISCUSSION ON RELATED WORK

**Closer comparisons with ILEED (Beliaev et al., 2022)**    ILEED is the closest work to ours, as both address imitation learning (IL) from multiple demonstrators with heterogeneous expertise. Despite the shared high-level motivation, the two approaches differ substantially in problem setting, modeling assumptions, methodologies and robustness properties, making them suitable for different practical regimes.

- **Problem setting.** ILEED considers the problem setting where each demonstrator's policy is a mixture of the optimal policy and a *uniformly random* policy, with the mixing coefficient interpreted as an expertise level.[5] While convenient, this "suboptimal = uniform noise" assumption is restrictive as real-world suboptimal behavior is typically structured, rather than near-uniform noise. For example, a demonstrator may consistently struggle with specific skills without acting randomly overall.

  In contrast, we relax this assumption by modeling each demonstrator's behavior as a mixture of the optimal policy and an *unknown* suboptimal policy. Our formulation imposes no uniformity or high-entropy constraints on the suboptimal component, enabling the method to capture realistic and highly structured forms of suboptimality.

- **Expertise modeling.** ILEED models expertise as a state-dependent function defined as a sigmoid of the inner product between the learned state and demonstrator embeddings. Although this formulation can represent state-varying expertise, it relies heavily on the quality of the recovered embeddings for *all* states visited by *all* demonstrators. In practice, when demonstrators explore only partial or disjoint regions of the state space, the resulting embeddings, and thus the inferred expertise levels, can become severely biased, causing significant policy degradation.

  Our method instead models expertise through the class-prior probability $\mathbb{P}(z=1)$, estimated via a surrogate scoring model that directly distinguishes optimal from suboptimal samples across demonstrators. This approach does not require each demonstrator to provide full coverage of the state space. Moreover, the trained scoring model outputs a sample-wise optimality score $\mathbb{P}(z=1 \mid s,a)$ that can be used as a reward to train an RL agent (SAC).

- **Methodology.** ILEED performs fully *offline* maximum-likelihood estimation of the policy, state embeddings, and demonstrator embeddings. As a result, it is sensitive to *distribution shift* between the demonstration distribution and the agent's on-policy distribution during training.

  Our method instead employs a *two-stage* procedure: we first learn a scoring model offline, then treat its log-outputs as a surrogate reward for an RL agent and *iteratively refine* the scorer using on-policy rollouts. To explicitly mitigate covariate shift, we incorporate mixup augmentation, an agent-matching penalty, and a relabeling mechanism, making our approach more robust when the agent visits states that rarely or never appear in the demonstrations.

- **Optimization stability.** ILEED jointly optimizes policy parameters, state embeddings, and demonstrator embeddings via a single maximum-likelihood objective. This fully coupled optimization can be unstable, as conflicting gradients among latent components may lead to degenerate solutions.

  Our method instead employs an EM-style alternating optimization for expertise estimation and scoring-model learning, for which we provide a convergence guarantee. Policy learning then proceeds through a standard RL pipeline using the scoring model as a reward proxy, yielding considerably more stable optimization behavior in practice.

Our empirical results (see Tab. 2, Fig. 3, Tab. 18) validate these distinctions. In particular, our method is advantageous when demonstrations contain structured suboptimality, exhibit low expertise levels, or when distribution shift between demonstrations and agent rollouts is significant.

---

[5]For continuous action spaces, ILEED replaces this uniform noise component with a high-entropy variant of the optimal policy. Specifically, the suboptimal component is modeled by scaling the variance of a Gaussian mixture policy, resulting in a near-uniform distribution rather than a structured suboptimal policy.

**Comparisons with T-REX/D-REX (Brown et al., 2019; 2020)** Both T-REX and D-REX learn a reward function from ranked demonstrations, allowing an agent to potentially outperform the demonstrator via extrapolation. T-REX relies on experts to provide ranking comparisons, while D-REX can automatically generate rankings by injecting noise into a cloned policy, based on the assumption that added noise leads to a continuous degradation in performance.

Compared to T-REX, our method does not require large amounts of supervision or expert ranking feedback during training. Moreover, while D-REX assumes homogeneous demonstrations and introduces heterogeneity artificially, our approach naturally handles heterogeneous datasets and is capable of inferring varying expertise levels within the data. Unlike D-REX, our method does not rely on the assumption that policy performance monotonically decreases with noise injection, which may not hold in more complex domains. In addition, both T-REX and D-REX first learn a surrogate reward function and then optimize a policy using reinforcement learning on this fixed reward, which can lead to covariate shift when the agent encounters states not present in the demonstrations. Our method addresses this issue by continuously fine-tuning the scoring model with new data collected during RL, resulting in improved robustness and generalization. Overall, our approach provides a flexible and practical framework for leveraging demonstrations, particularly in settings with heterogeneous data and limited supervision.

**Comparisons with VPIL (Cai et al., 2023)** Vaguely Pairwise Imitation Learning (VPIL) investigates a weak-supervision setting in which the annotator provides vague pairwise feedback: a pair of demonstrations is labeled only when their quality differs substantially, resulting in two contaminated datasets that still contain mixtures of expert and non-expert trajectories. VPIL leverages this pairwise structure to recover the expert occupancy measure through mixture-proportion estimation (MPE) (Ramaswamy et al., 2016) and risk rewriting, and its COMPILER-E algorithms integrate the recovered distribution into a GAIL-style adversarial imitation framework.

Our problem setting is more general and arguably more challenging. Unlike VPIL, which assumes access to vague pairwise comparison labels, we consider a scenario in which the agent receives only heterogeneous demonstrations of unknown and varying quality, with no preference, ranking, or optimality labels. In the absence of such supervision, the learner must infer both the underlying demonstration quality and the reward surrogate directly from unlabeled trajectories.

The methodological differences therefore extend beyond supervision assumptions. Rather than reconstructing an expert occupancy measure from pairwise datasets, we learn a fine-grained state–action optimality scoring model through a surrogate demonstrator classification objective and an EM-style latent expertise level estimation procedure. The trained scoring model then serves as a surrogate reward within an iterative policy-learning loop, enabling robust imitation learning directly from heterogeneous, unlabeled demonstrators without relying on any preference-based supervision.

**Comparisons with AIRL (Kingma & Ba, 2018)** Adversarial Inverse Reinforcement Learning (AIRL) extends the GAIL framework by structuring the discriminator to decompose into a state-only reward term and a shaping term, enabling recovery of a reward function that is disentangled from environment dynamics. This structure allows AIRL to learn portable rewards that can be re-optimized in new or modified environments.

Our problem setting, however, is fundamentally different. Rather than assuming demonstrations come from a single (near-optimal) expert, we address imitation learning from heterogeneous, unlabeled, and potentially highly suboptimal demonstrators with unknown expertise levels. Although our method learns a scoring model that serves as a surrogate reward, the objective is not reward recovery or transfer, but robust policy learning when expert dominance does not hold. AIRL, like GAIL, is not designed to handle heterogeneous demonstrators and presumes access to expert demonstrations. Therefore, their assumptions and objectives cannot address the key challenges in our setting.

## B  SURROGATE DEMONSTRATOR CLASSIFICATION TRAINING

We provide more details about the algorithm and theory of Surrogate Demonstrator Classification training.

---

**Algorithm 3** Learning scoring model

---

1: **Input**: $\{(i, D_i)\}_{i=1}^m, \{\alpha_i\}_{i=1}^m$ , and $\alpha'$
2: **Initialization**: scoring model $f_\phi(s, a)$
3: Compute of $T(\cdot) = [T_1(\cdot), \cdots, T_m(\cdot)]$ using Eq. 3 based on $\{\alpha_i\}_{i=1}^m$ and $\alpha'$
4: **for** $t = 1, 2, \cdots$ **do**
5:     Sample examples: $\bar{z}, (s, a) \sim \{(i, D_i)\}_{i=1}^m$
6:     Update $f_\phi$ by minimizing the empirical risk from Eq. 4
7: **end for**
8: **Return** $f_\phi$

---

We consider demonstrator heterogeneity, i.e., there $\exists\, i, j \in [m], i \neq j$, such that $\alpha_i \neq \alpha_j$. According to Lu et al. (2021), $f_{\phi^*_{\text{surr}}}$ that recovered by removing transformation layer $T(\cdot)$ of the optimal surrogate classifier $\overline{f}_{\phi^*}$ (the minimizer of $\mathbb{E}_{(s,a)\sim\rho}[\ell(\overline{f}_\phi(s, a), \bar{z})] = \mathbb{E}_{(s,a)\sim\rho}[\ell(T(f_\phi(s, a)), \bar{z})]$) is identical to the optimal classifier $f_{\phi^*}$ that minimize the risk $\mathbb{E}_{(s,a)\sim\rho}[\ell(f_\phi(s, a), z)]$, which is not possible if the heterogeneity setting is invalid. Thus, through the transformation function transformation $T(\cdot)$, the optimality scoring model $f_{\phi^*}$ can be learned by minimizing the surrogate classification risk in Eq. 4.

## C  PROOF OF THEOREM 2

*Proof.* Let $c = \mathbb{P}(z = 1)$ be the optimality prior probability in the dataset. By Bayes' rule, we have

$$f(s, a) = \mathbb{P}(z = 1 \mid s, a) = c\frac{\rho^{\pi^*}(s, a)}{\rho(s, a)}.$$

Thus, the surrogate RL objective can be written as

$$J_f(\pi) = \sum_{s,a} \rho^\pi(s, a)f_\phi(s, a) = c\sum_{s,a} \frac{\rho^\pi(s, a)\rho^{\pi^*}(s, a)}{\rho(s, a)}.$$

Define the following vectors indexed by $(s, a)$: $x_{s,a} = \frac{\rho^\pi(s,a)}{\sqrt{\rho(s,a)}}$ and $y_{s,a} = \frac{\rho^{\pi^*}(s,a)}{\sqrt{\rho(s,a)}}$, then we have

$$
\begin{aligned}
J_f(\pi) &= c\langle x, y\rangle \\
&\leq c\|x\|_2\|y\|_2 \\
&= c\sqrt{\sum_{s,a} \frac{\rho^\pi(s, a)^2}{\rho(s, a)}}\sqrt{\sum_{s,a} \frac{\rho^{\pi^*}(s, a)^2}{\rho(s, a)}},
\end{aligned}
\tag{7}
$$

where we applied the Cauchy-Schwarz inequality in (7). The above equality holds if and only if $x = \beta y$ for some scalar $\beta \geq 0$, that is,

$$\frac{\rho^\pi(s, a)}{\sqrt{\rho(s, a)}} = \beta\frac{\rho^{\pi^*}(s, a)}{\sqrt{\rho(s, a)}}, \ \forall (s, a) \text{ in supp}(\rho^\pi).$$

Since $\sqrt{\rho(s, a)} > 0$, the above equality reduces to

$$\rho^\pi(s, a) = \beta\rho^{\pi^*}(s, a), \ \forall (s, a) \text{ in supp}(\rho^\pi).$$

Summing both sides over $(s, a)$, and given the fact that both $\rho^\pi$ and $\rho^{\pi^*}$ have the same total mass $\sum_{s,a} \rho^\pi(s, a) = \sum_{t=1}^T \gamma^t > 0$, we obtain that $\beta = 1$, and therefore $\rho^\pi = \rho^{\pi^*}$. This proves the optimal occupancy recovery.

If $\text{supp}(\rho^{\pi^*}) \subseteq \text{supp}(\rho)$, then for any state $s$ with $\rho^{\pi^*}(s) > 0$, we have $\rho^{\hat{\pi}^*}(s) > 0$. The conditional action probabilities are then recovered by normalization

$$\hat{\pi}^*(a \mid s) = \frac{\rho^{\hat{\pi}^*}(s, a)}{\rho^{\hat{\pi}^*}(s)} = \frac{\rho^{\pi^*}(s, a)}{\rho^{\pi^*}(s)} = \pi^*(a \mid s),$$

where the second equality is due to $\rho^{\hat{\pi}^*} = \rho^{\pi^*}$. Thus $\hat{\pi}^*$ matches $\pi^*$ for every state $s$ in the support of $\rho^{\pi^*}$. $\qquad\square$

## D    PROOF OF THEOREM 3

*Proof.* We first prove that the sequence $\{\mathcal{L}(\phi^t, \alpha^t)\}$ is non-increasing. If $\mathcal{L}(\phi, \alpha)$ is lower-bounded (e.g., by zero in many loss formulations), then by the monotone convergence theorem, the sequence must converge.

$$\mathcal{L}(\phi, \alpha) = \mathbb{E}_{(s,a) \sim \rho} \left[ \ell(T_\alpha(f_\phi(s, a)), \bar{z}) \right] \tag{8}$$

Minimizing the cross-entropy loss in Eq. 8 is equivalent to maximizing a log likelihood (Shangnan & Wang, 2021). We define a log likelihood function through the conditional distribution $p(\bar{z}|(s, a))$ in the form of discriminative training. With scoring model parameters and hidden parameters of expertise levels, the log likelihood ($ll$) can be formulated as

$$ll(\phi, \alpha) := \ln p(\bar{z}|(s, a), \phi) = \sum_\alpha p(\alpha) \ln p(\bar{z}|(s, a), \phi) \tag{9}$$

To prove that $\{\mathcal{L}(\phi^t, \alpha^t)\}$ is monotonically non-increasing, we only have to prove $\{ll(\phi^t, \alpha^t)\}$ is monotonically non-decreasing, i.e., $ll(\phi^{t+1}, \alpha^{t+1}) \geq ll(\phi^t, \alpha^t)$.

We define $L(q, \phi) := \sum_\alpha q(\alpha) \ln \frac{p(\bar{z}, \alpha|(s,a), \phi)}{q(\alpha)}$, $q := q(\alpha)$, and $p := p(\alpha|\bar{z}, (s, a), \phi)$

$$
\begin{aligned}
L(q, \phi) &= \sum_\alpha q(\alpha) \ln \frac{p(\bar{z}, \alpha|(s, a), \phi)}{q(\alpha)} \\
&= \sum_\alpha q(\alpha) \left[ \ln p(\alpha|\bar{z}, (s, a), \phi) + \ln p(\bar{z}|(s, a), \phi) - \ln q(\alpha) \right] \\
&= \sum_\alpha q(\alpha) \ln \frac{p(\alpha|\bar{z}, (s, a), \phi)}{q(\alpha)} + \sum_\alpha q(\alpha) \ln p(\bar{z}|(s, a), \phi) \\
&= -KL(q||p) + \ln p(\bar{z}|(s, a), \phi)
\end{aligned}
\tag{10}
$$

Then, the likelihood $ll(\phi, \alpha)$ can be represented as

$$
\begin{aligned}
\ln p(d|(s, a), \phi) &= L(q, \phi) + KL(q||p), \\
&\geq L(q, \phi)
\end{aligned}
\tag{11}
$$

Since $KL(q||p) \geq 0$, $L(q, \phi)$ is the lower bound of $ll(\phi, \alpha)$.

For likelihood function $ll(\phi^t, \alpha^t)$ at iteration $t$,

$$
\begin{aligned}
ll(\phi^t, \alpha^t) &= \ln p(\bar{z}|(s, a), \phi^t) \\
&= L(q(\alpha^t), \phi^t) + KL(q(\alpha^t)||p(\alpha|\bar{z}, (s, a), \phi^t))
\end{aligned}
\tag{12}
$$

Since $L(q(\alpha^t), \phi^t) = \ln p(\bar{z}|(s, a), \phi^t) - KL(q(\alpha^t)||p(\alpha|\bar{z}, (s, a), \phi^t))$ is the lower bound $ll(\phi^t, \alpha^t)$. We update $\alpha^t$ by letting $q(\alpha^t) = p(\alpha|\bar{z}, (s, a), \phi^t)$ to maximize the lower bound, where $p(\alpha|\bar{z}, (s, a), \phi^t)$ can be calculated based on the second equation in Eq.5. Thus, $KL(q(\alpha^{t+1})||p(\alpha|\bar{z}, (s, a), \phi^t)) = 0$. Thus, $ll(\phi^t, \alpha^{t+1})$ can be expressed as

$$
\begin{aligned}
ll(\phi^t, \alpha^{t+1}) &= \ln p(\bar{z}|(s, a), \phi^t) \\
&= L(q(\alpha^{t+1}), \phi^t) + KL(q(\alpha^{t+1})||p(\alpha|\bar{z}, (s, a), \phi^t)) \\
&= L(q(\alpha^{t+1}), \phi^t)
\end{aligned}
\tag{13}
$$

By updating $\alpha$, we increase the lower bound of $ll(\phi^t, \alpha^{t+1})$ equals to $\ln p(\bar{z}|(s, a), \phi^t)$. Then we maximize $ll(\phi^t, \alpha^{t+1})$ with regard to $\phi$ using a gradient decent method, $ll(\phi^{t+1}, \alpha^{t+1})$ must be not smaller than $ll(\phi^t, \alpha^{t+1})$. Finally, we have $ll(\phi^{t+1}, \alpha^{t+1}) \geq ll(\phi^t, \alpha^{t+1}) = \ln p(\bar{z}|(s, a), \phi^t) = ll(\phi^t, \alpha^t)$

□

# E   EXPERIMENT DETAILS

## E.1   DETAILED RESULTS

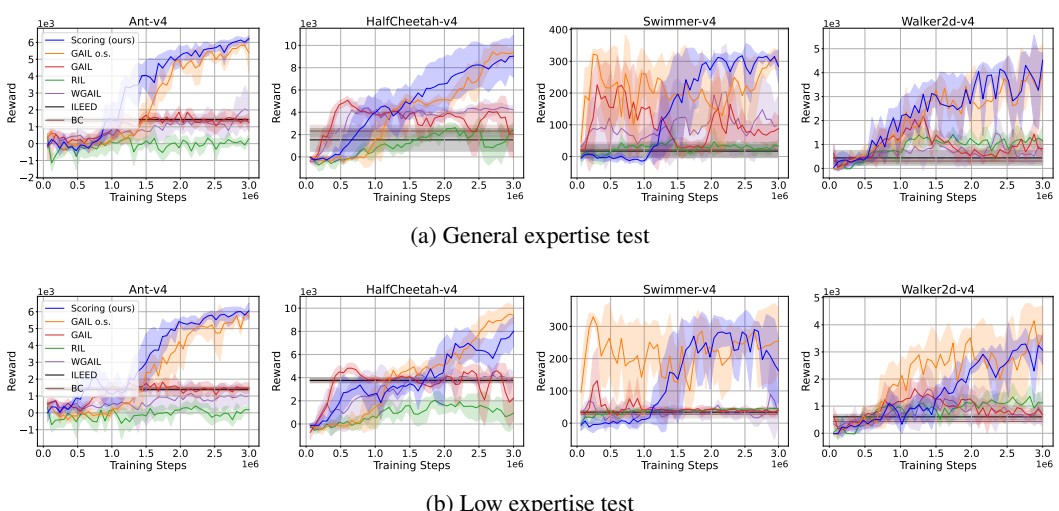

(a) General expertise test

(b) Low expertise test

Figure 3: Performance comparison (mean ± std, shown as shaded region) with baselines. Supplementary results for Sec. 4.2.

Table 5: Performance (reward) of checkpoints used as optimal and suboptimal policies across environments. Values are reported as mean ± standard deviation.

| Environment | Optimal policy $\pi^*$ | Suboptimal policy $\pi^{sub}$ |
|---|---|---|
| Ant-v4 | $6766.26 \pm 109.70$ | $1266.37 \pm 533.96$ |
| HalfCheetah-v4 | $15947.53 \pm 39.08$ | $4030.23 \pm 169.82$ |
| Swimmer-v4 | $339.80 \pm 1.15$ | $41.60 \pm 2.34$ |
| Walker-v4 | $5638.94 \pm 49.76$ | $554.38 \pm 267.99$ |

Table 6: True expertise levels $\{\alpha_i\}_{i=1}^m$ for different numbers of demonstrators across two tests. Expertise levels are uniformly distributed within each range.

| Demonstrator | 1 | 2 | 3 | 4 | 5 | 6 | 7 | 8 |
|---|---|---|---|---|---|---|---|---|
| **General expertise test** | | | | | | | | |
| 2 demonstrators | 0.100 | 0.900 | – | – | – | – | – | – |
| 4 demonstrators | 0.100 | 0.366 | 0.633 | 0.900 | – | – | – | – |
| 6 demonstrators | 0.100 | 0.260 | 0.420 | 0.580 | 0.740 | 0.900 | – | – |
| 8 demonstrators | 0.100 | 0.214 | 0.329 | 0.443 | 0.557 | 0.671 | 0.786 | 0.900 |
| **Low expertise test** | | | | | | | | |
| 2 demonstrators | 0.050 | 0.150 | – | – | – | – | – | – |
| 4 demonstrators | 0.050 | 0.083 | 0.117 | 0.150 | – | – | – | – |
| 6 demonstrators | 0.050 | 0.070 | 0.090 | 0.110 | 0.130 | 0.150 | – | – |
| 8 demonstrators | 0.050 | 0.064 | 0.079 | 0.093 | 0.107 | 0.121 | 0.136 | 0.150 |

Table 7: Estimated expertise levels $\{\hat{\alpha}_i\}_{i=1}^m$, showing the mean estimate and the mean $\pm$ standard deviation of error across five random seeds. True values are shown separately. Supplementary results for Sec. 4.3

| Demonstrator | 1 | 2 | 3 | 4 | 5 | 6 | Mean error $\pm$ std |
|---|---|---|---|---|---|---|---|
| **General expertise test** | | | | | | | |
| Ant-v4 | 0.100 | 0.260 | 0.420 | 0.580 | 0.740 | 0.900 | $(2.67 \pm 1.28) \times 10^{-4}$ |
| HalfCheetah-v4 | 0.100 | 0.260 | 0.420 | 0.580 | 0.740 | 0.900 | $(2.00 \pm 2.00) \times 10^{-5}$ |
| Hopper-v4 | 0.101 | 0.259 | 0.418 | 0.575 | 0.734 | 0.892 | $(3.83 \pm 2.73) \times 10^{-3}$ |
| Swimmer-v4 | 0.100 | 0.260 | 0.420 | 0.580 | 0.740 | 0.900 | $(2.00 \pm 2.00) \times 10^{-5}$ |
| Walker2d-v4 | 0.100 | 0.260 | 0.420 | 0.581 | 0.741 | 0.900 | $(3.27 \pm 3.12) \times 10^{-4}$ |
| True expertise levels | 0.100 | 0.260 | 0.420 | 0.580 | 0.740 | 0.900 | |
| **Low expertise test** | | | | | | | |
| Ant-v4 | 0.148 | 0.170 | 0.187 | 0.207 | 0.222 | 0.246 | $(9.65 \pm 0.24) \times 10^{-2}$ |
| HalfCheetah-v4 | 0.086 | 0.105 | 0.127 | 0.146 | 0.167 | 0.190 | $(3.67 \pm 0.15) \times 10^{-2}$ |
| Hopper-v4 | 0.156 | 0.176 | 0.195 | 0.216 | 0.231 | 0.257 | $(1.05 \pm 0.02) \times 10^{-1}$ |
| Swimmer-v4 | 0.077 | 0.100 | 0.118 | 0.139 | 0.158 | 0.177 | $(2.81 \pm 0.11) \times 10^{-2}$ |
| Walker2d-v4 | 0.179 | 0.206 | 0.219 | 0.248 | 0.263 | 0.287 | $(1.34 \pm 0.04) \times 10^{-1}$ |
| True expertise levels | 0.050 | 0.070 | 0.090 | 0.110 | 0.130 | 0.150 | |

Table 8: Classification / Labeling results (mean $\pm$ std). Stage 1: labeling using the pretrained scoring model. Stage 2: relabeling using the fine-tuned scoring model. TP (true positive), TN (true negative), FP (false positive), FN (false negative). Improvement (%) shows the Stage 1 $\rightarrow$ Stage 2 percentage change of accuracy and precision after relabeling. Supplementary results for Sec. 4.4

| Environment | Stage | TP | TN | FP | FN | Accuracy / Precision | Improvement (%) |
|---|---|---|---|---|---|---|---|
| **General expertise test** | | | | | | | |
| Ant-v4 | 1 | $14988.6 \pm 3.9$ | $14984.2 \pm 10.1$ | $15.8 \pm 10.1$ | $11.4 \pm 3.9$ | 0.9991 / 0.9989 | - |
| | 2 | $14984.0 \pm 5.5$ | $14984.0 \pm 5.5$ | $16.0 \pm 5.5$ | $16.0 \pm 5.5$ | 0.9989 / 0.9989 | -0.02 / 0.00 |
| HalfCheetah-v4 | 1 | $15000.0 \pm 0.0$ | $14999.4 \pm 0.8$ | $0.6 \pm 0.8$ | $0.0 \pm 0.0$ | 1.0000 / 1.0000 | - |
| | 2 | $15000.0 \pm 0.0$ | $15000.0 \pm 0.0$ | $0.0 \pm 0.0$ | $0.0 \pm 0.0$ | 1.0000 / 1.0000 | 0.00 / 0.00 |
| Swimmer-v4 | 1 | $15000.0 \pm 0.0$ | $14999.4 \pm 1.2$ | $0.6 \pm 1.2$ | $0.0 \pm 0.0$ | 1.0000 / 1.0000 | - |
| | 2 | $14991.4 \pm 16.2$ | $14991.4 \pm 16.2$ | $8.6 \pm 16.2$ | $8.6 \pm 16.2$ | 0.9994 / 0.9994 | -0.06 / -0.06 |
| Walker2d-v4 | 1 | $14992.0 \pm 2.2$ | $14985.0 \pm 6.1$ | $15.0 \pm 6.1$ | $8.0 \pm 2.2$ | 0.9992 / 0.9990 | - |
| | 2 | $14990.4 \pm 1.6$ | $14990.4 \pm 1.6$ | $9.6 \pm 1.6$ | $9.6 \pm 1.6$ | 0.9994 / 0.9994 | 0.02 / 0.04 |
| **Low expertise test** | | | | | | | |
| Ant-v4 | 1 | $2997.6 \pm 1.5$ | $24101.4 \pm 735.5$ | $2898.6 \pm 735.5$ | $2.4 \pm 1.5$ | 0.9033 / 0.5178 | - |
| | 2 | $2947.0 \pm 32.9$ | $26947.0 \pm 32.9$ | $53.0 \pm 32.9$ | $53.0 \pm 32.9$ | 0.9965 / 0.9823 | 10.32 / 89.71 |
| HalfCheetah-v4 | 1 | $3000.0 \pm 0.0$ | $25898.2 \pm 355.8$ | $1101.8 \pm 355.8$ | $0.0 \pm 0.0$ | 0.9633 / 0.7368 | - |
| | 2 | $2997.4 \pm 5.2$ | $26997.4 \pm 5.2$ | $2.6 \pm 5.2$ | $2.6 \pm 5.2$ | 0.9998 / 0.9991 | 3.79 / 35.60 |
| Swimmer-v4 | 1 | $3000.0 \pm 0.0$ | $26156.2 \pm 393.5$ | $843.8 \pm 393.5$ | $0.0 \pm 0.0$ | 0.9719 / 0.7884 | - |
| | 2 | $2637.2 \pm 290.4$ | $26637.2 \pm 290.4$ | $362.8 \pm 290.4$ | $362.8 \pm 290.4$ | 0.9758 / 0.8791 | 0.40 / 11.50 |
| Walker2d-v4 | 1 | $2965.0 \pm 42.7$ | $22953.2 \pm 890.1$ | $4046.8 \pm 890.1$ | $35.0 \pm 42.7$ | 0.8639 / 0.4301 | - |
| | 2 | $2772.0 \pm 170.3$ | $26772.0 \pm 170.3$ | $228.0 \pm 170.3$ | $228.0 \pm 170.3$ | 0.9848 / 0.9240 | 13.99 / 114.83 |

Table 9: Ablation results on the impact of relabeling and early stopping in Stage 2 on Ant-v4 task. Reported as mean $\pm$ std of reward across random seeds; each seed's value is the mean over its last five checkpoints. Supplementary results for Sec. 4.5

| Method | Expertise $[0.1 - 0.9]$ | Expertise $[0.05 - 0.15]$ |
|---|---|---|
| Scoring (ours) | $\mathbf{6084.6 \pm 146.9}$ | $\mathbf{5932.0 \pm 335.2}$ |
| Scoring (no relabel) | $5593.0 \pm 564.2$ | $3316.2 \pm 1207.8$ |
| Scoring (no early stop) | $2129.8 \pm 2205.8$ | $1018.2 \pm 250.0$ |

Table 10: Ablation study on top-k hyperparameter on Ant-v4 task. Reported as mean $\pm$ std of reward across random seeds; each seed's value is the mean over its last five checkpoints.. Supplementary results for Sec. 4.5

| Top-k fraction | Expertise $[0.1 - 0.9]$ | Expertise $[0.05 - 0.15]$ |
|---|---|---|
| 0.05 | – | $5887.6 \pm 263.8$ |
| 0.1 | $5403.8 \pm 378.7$ | $\mathbf{5932.0 \pm 335.2}$ |
| 0.15 | – | $3663.4 \pm 1950.9$ |
| 0.2 | – | $1756.6 \pm 902.6$ |
| 0.3 | $5809.0 \pm 274.8$ | – |
| 0.5 | $\mathbf{6084.6 \pm 146.9}$ | – |
| 0.55 | $4109.2 \pm 2204.9$ | – |
| 0.6 | $3641.0 \pm 1237.7$ | – |
| 0.7 | $2383.0 \pm 1059.7$ | – |

Table 11: Estimated expertise levels $\{\hat{\alpha}_i\}_{i=1}^m$ on Ant-v4 task, showing the mean estimate and the mean $\pm$ standard deviation of error across five random seeds. True values are shown separately. Results shown for different numbers of demonstrators (2, 4, 6, 8). Supplementary results for Sec. 4.5

| Demonstrator | Type | 1 | 2 | 3 | 4 | 5 | 6 | 7 | 8 | Mean error $\pm$ std |
|---|---|---|---|---|---|---|---|---|---|---|
| **General expertise test** | | | | | | | | | | |
| 2 demonstrators | Est. | 0.101 | 0.899 | - | - | - | - | - | - | $(9.60 \pm 1.60) \times 10^{-4}$ |
| | True | 0.100 | 0.900 | - | - | - | - | - | - | - |
| 4 demonstrators | Est. | 0.100 | 0.366 | 0.633 | 0.899 | - | - | - | - | $(4.50 \pm 2.96) \times 10^{-4}$ |
| | True | 0.100 | 0.367 | 0.633 | 0.900 | - | - | - | - | - |
| 6 demonstrators | Est. | 0.100 | 0.260 | 0.420 | 0.580 | 0.740 | 0.900 | - | - | $(2.67 \pm 1.28) \times 10^{-4}$ |
| | True | 0.100 | 0.260 | 0.420 | 0.580 | 0.740 | 0.900 | - | - | - |
| 8 demonstrators | Est. | 0.100 | 0.214 | 0.329 | 0.443 | 0.557 | 0.671 | 0.785 | 0.899 | $(2.50 \pm 1.97) \times 10^{-4}$ |
| | True | 0.100 | 0.214 | 0.329 | 0.443 | 0.557 | 0.671 | 0.786 | 0.900 | - |
| **Low expertise test** | | | | | | | | | | |
| 2 demonstrators | Est. | 0.193 | 0.287 | - | - | - | - | - | - | $(1.40 \pm 0.03) \times 10^{-1}$ |
| | True | 0.050 | 0.150 | - | - | - | - | - | - | - |
| 4 demonstrators | Est. | 0.106 | 0.143 | 0.173 | 0.208 | - | - | - | - | $(5.77 \pm 0.16) \times 10^{-2}$ |
| | True | 0.050 | 0.083 | 0.117 | 0.150 | - | - | - | - | - |
| 6 demonstrators | Est. | 0.148 | 0.170 | 0.187 | 0.207 | 0.222 | 0.246 | - | - | $(9.65 \pm 0.24) \times 10^{-2}$ |
| | True | 0.050 | 0.070 | 0.090 | 0.110 | 0.130 | 0.150 | - | - | - |
| 8 demonstrators | Est. | 0.073 | 0.087 | 0.101 | 0.115 | 0.130 | 0.144 | 0.160 | 0.175 | $(2.31 \pm 0.10) \times 10^{-2}$ |
| | True | 0.050 | 0.064 | 0.079 | 0.093 | 0.107 | 0.121 | 0.136 | 0.150 | - |

Table 12: Ablation on the different number of demonstrators for Ant-v4 task. Reported numbers are mean $\pm$ std across random seeds; each seed's value is the mean over its last five checkpoints. Supplementary results for Sec. 4.5

| Number of Demonstrators | Expertise $[0.1 - 0.9]$ | Expertise $[0.05 - 0.15]$ |
|---|---|---|
| 2 | $5634.0 \pm 617.8$ | $6050.6 \pm 120.4$ |
| 4 | $5934.0 \pm 264.3$ | $5637.2 \pm 772.5$ |
| 6 | $\mathbf{6084.6 \pm 146.9}$ | $5932.0 \pm 335.2$ |
| 8 | $5735.0 \pm 225.6$ | $\mathbf{6146.6 \pm 107.2}$ |

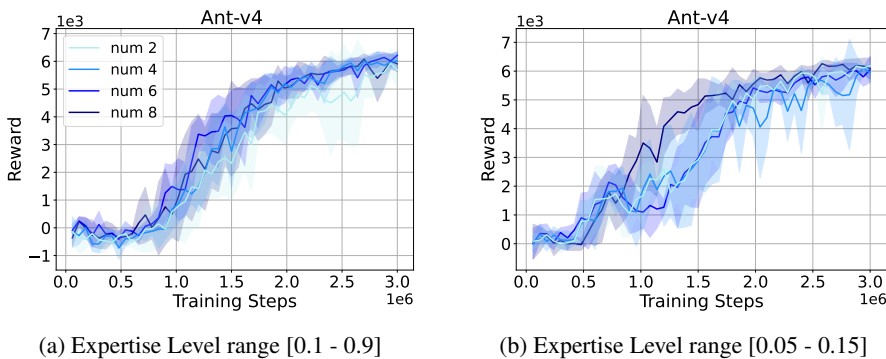

(a) Expertise Level range [0.1 - 0.9]    (b) Expertise Level range [0.05 - 0.15]

Figure 4: Ablation on the different number of demonstrators for Ant-v4 task (mean $\pm$ std, shown as shaded region). Supplementary results for Sec. 4.5

**Number of Demonstrators:** The results in Tab. 11 show that the estimation error of expertise levels in low expertise test increase when the number of demonstrators is reduced. However, the reward obtained by the IL agent in Stage 2 (see Tab. 12) does not decrease substantially. These results demonstrate the robustness of our method with respect to varying demonstrator counts in terms of IL performance.

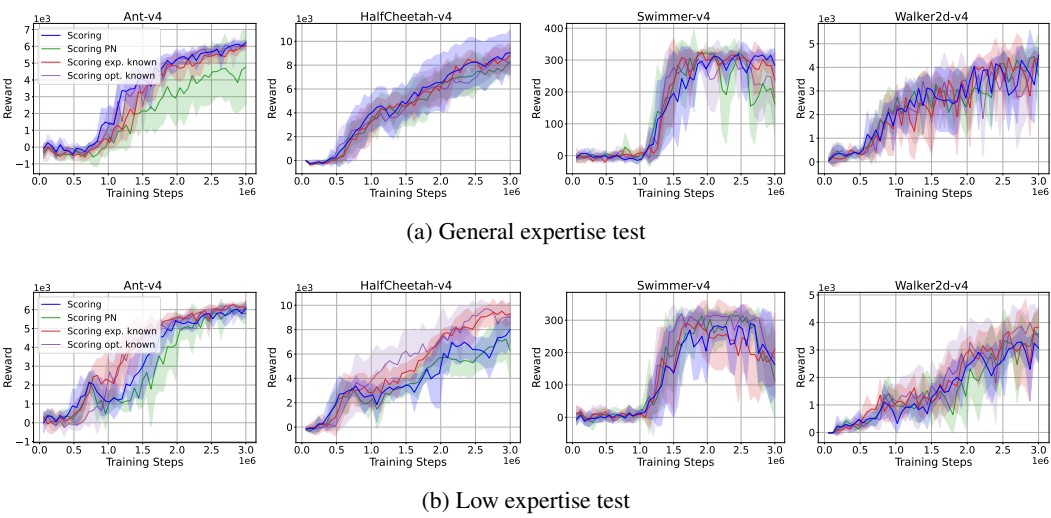

(a) General expertise test

(b) Low expertise test

Figure 5: Comparative analysis of our method and its variants.

### E.2 ADDITIONAL ANALYSIS OF METHOD VARIANTS

We analyze our algorithm by comparing it against several variants under different settings. (1) **Scoring**: original method; (2) **Scoring PN**: in Stage 2, the first term $\widehat{\mathbb{E}}_{(s,a)\sim\rho}\big[\mathcal{L}(f_\phi(s,a)\big]$ (demon-

Table 13: Classification results of the scoring model on imperfect demonstrations (Ant-v4) for Stage 1 and Stage 2 (Accuracy / Precision, %), with Stage 1 → Stage 2 relative improvement (%). Ablation study about expert query count. Supplementary results see Fig. 6

| Test | Stage | Query 1 | Impr. | Query 3 | Impr. | Query 5 | Impr. | Query 7 | Impr. | Query 9 | Impr. |
|------|-------|---------|-------|---------|-------|---------|-------|---------|-------|---------|-------|
| General | 1 | 99.93 / 99.97 | - | 99.92 / 99.94 | - | 99.91 / 99.89 | - | 99.92 / 99.96 | - | 99.94 / 99.95 | - |
| Expertise | 2 | 99.92 / 99.92 | -0.01 / -0.05 | 99.90 / 99.90 | -0.02 / -0.04 | 99.89 / 99.89 | -0.02 / 0.00 | 99.89 / 99.89 | -0.03 / -0.07 | 99.91 / 99.91 | -0.03 / -0.04 |
| Low | 1 | 96.26 / 75.31 | - | 97.33 / 80.17 | - | 90.33 / 51.78 | - | 97.02 / 78.81 | - | 97.14 / 79.26 | - |
| Expertise | 2 | 99.64 / 98.22 | 3.51 / 30.42 | 99.52 / 97.60 | 2.25 / 21.74 | 99.65 / 98.23 | 10.32 / 89.71 | 99.48 / 97.40 | 2.54 / 23.59 | 99.73 / 98.65 | 2.67 / 24.46 |

stration discriminative loss) in Eq. 6 is replaced with a PN loss $\widehat{\mathbb{E}}_{\substack{(s,a)\sim\hat{\rho}^{\pi^*} \\ (s',a')\sim\hat{\rho}^{\pi^{\mathrm{sub}}}}} \left[ \mathcal{L}_{PN}(f_\phi(s,a,s',a')) \right]$,

where $\mathcal{L}_{PN}(f_\phi(s,a,s',a') = \log\left(1 - f_\phi(s,a)\right) + \log\left(f_\phi(s',a')\right))$, and $\pi^{\mathrm{sub}}$ represent the pseudo-suboptimal data inferred from $D_{\mathrm{subopt}} = D \setminus D_{\mathrm{opt}}$. (3) **Scoring exp. known**: the true expertise levels are assumed to be given in both Stage 1 and 2. (4) **Scoring opt. known**: in Stage 2, $D_{\mathrm{opt}}$ is provided based on the true labels.

The results in Fig. 5 show that, overall, all methods achieve comparable performance. **Scoring PN** yields slightly lower rewards in general expertise test on Ant and Swimmer tasks, which may be attributed to the overconfidence issue introduced by self-labeling when using the PN loss. This result highlights the effectiveness of our demonstration discriminative loss. **Scoring exp. known** and **Scoring opt. known** serve as oracle variants (with oracle knowledge of expertise levels and optimality labels, respectively) and achieve only marginally better performance than the original method in the low expertise test on HalfCheetah and Walker2d. This observation highlights the effectiveness of our approach in expertise-level estimation and optimality prediction when learning from imperfect demonstrations.

Table 14: Ablation study on Query times for Ant-v4. Reported as mean ± std of reward across random seeds; each seed's value is the mean over its last five checkpoints.

| Query times | Expertise [0.1 − 0.9] | Expertise [0.05 − 0.15] |
|-------------|----------------------|-------------------------|
| 1 | $5433.6 \pm 365.6$ | $5810.2 \pm 301.3$ |
| 3 | $5655.2 \pm 307.3$ | $6011.2 \pm 219.9$ |
| 5 | $6084.6 \pm 146.9$ | $5932.0 \pm 335.2$ |
| 7 | $5981.4 \pm 264.6$ | $5944.2 \pm 279.8$ |
| 9 | $5881.6 \pm 265.3$ | $6130.8 \pm 126.8$ |

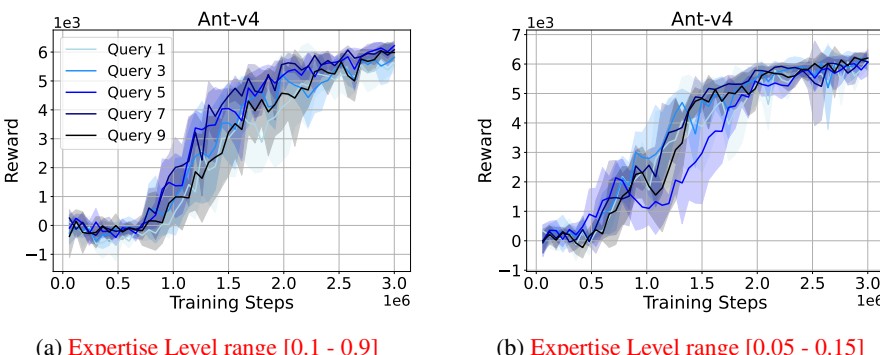

(a) Expertise Level range [0.1 - 0.9]          (b) Expertise Level range [0.05 - 0.15]

Figure 6: Ablation on the different expert query count for Ant-v4 task (mean ± std, shown as shaded region). Supplementary results see Tab. 13

### E.3 REPRODUCIBILITY STATEMENT

To ensure the reproducibility of this work, details of the training parameters are provided in Tab. 21. All reported results are averaged over five random seeds (0–4) per condition/method, with mean and standard deviation reported. The corresponding code will be released as open-source upon publication of this work.

Table 15: Classification results of the scoring model on imperfect demonstrations for Stage 1 - Accuracy and Precision (%, mean $\pm$ std) - for Ant-v4 across different $\alpha'$ values (expected expertise in the target environment). Supplementary results see Fig. 7

| $\alpha'$ | General expertise level | | Low expertise level | |
|---|---|---|---|---|
| | **Accuracy** | **Precision** | **Accuracy** | **Precision** |
| 0.1 | $59.97 \pm 19.94$ | $20.00 \pm 39.99$ | $90.00 \pm 0.00$ | $0.00 \pm 0.00$ |
| 0.2 | $89.94 \pm 19.97$ | $79.97 \pm 39.99$ | $90.00 \pm 0.00$ | $0.00 \pm 0.00$ |
| 0.3 | $99.92 \pm 0.02$ | $99.96 \pm 0.03$ | - | - |
| 0.4 | $99.94 \pm 0.01$ | $99.98 \pm 0.02$ | $90.00 \pm 0.00$ | $0.00 \pm 0.00$ |
| 0.45 | - | - | $96.44 \pm 2.43$ | $88.46 \pm 11.92$ |
| 0.5 | $99.91 \pm 0.03$ | $99.89 \pm 0.07$ | $90.33 \pm 2.45$ | $51.78 \pm 7.56$ |
| 0.55 | - | - | $81.07 \pm 35.54$ | $77.18 \pm 34.12$ |
| 0.6 | $99.92 \pm 0.02$ | $99.90 \pm 0.05$ | $20.03 \pm 20.07$ | $12.01 \pm 4.02$ |
| 0.7 | $99.93 \pm 0.02$ | $99.93 \pm 0.05$ | $27.21 \pm 34.42$ | $22.35 \pm 12.70$ |
| 0.8 | $99.92 \pm 0.02$ | $99.90 \pm 0.04$ | - | - |
| 0.9 | $79.91 \pm 24.43$ | $79.87 \pm 24.39$ | $10.00 \pm 0.00$ | $10.00 \pm 0.00$ |

Table 16: Ablation study on $\alpha'$ for Ant-v4. Reported as mean $\pm$ std of reward across random seeds; each seed's value is the mean over its last five checkpoints. Note that there are no policy learning results appear for expertise $[0.05 - 0.15]$ when $\alpha' \leq 0.4$ because all imperfect demonstrations are labeled sub-optimal, leaving the optimal set empty.

| $\alpha'$ value | Expertise $[0.1 - 0.9]$ | Expertise $[0.05 - 0.15]$ |
|---|---|---|
| 0.1 | $1149.4 \pm 2298.8$ | – |
| 0.2 | $3700.6 \pm 2286.5$ | – |
| 0.3 | $5241.6 \pm 957.9$ | – |
| 0.4 | $5308.0 \pm 377.7$ | – |
| 0.45 | – | $5509.0 \pm 493.4$ |
| 0.5 | $6084.6 \pm 146.9$ | $5932.0 \pm 335.2$ |
| 0.55 | – | $5496.6 \pm 509.1$ |
| 0.6 | $5579.0 \pm 416.2$ | $2927.8 \pm 2064.2$ |
| 0.7 | $5542.8 \pm 261.6$ | $3497.6 \pm 2349.8$ |
| 0.8 | $5262.6 \pm 330.8$ | – |
| 0.9 | $5090.2 \pm 573.1$ | $3171.0 \pm 1910.6$ |

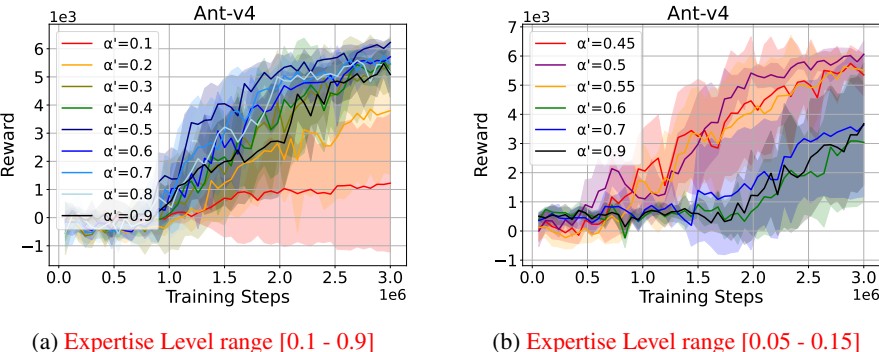

(a) Expertise Level range [0.1 - 0.9]          (b) Expertise Level range [0.05 - 0.15]

Figure 7: Ablation on the different $\alpha'$ values (expected expertise in the target environment) for Ant-v4 task (mean $\pm$ std, shown as shaded region). Supplementary results see Tab. 15

| Env | General Expertise (Mean $\pm$ Std) | Low Expertise (Mean $\pm$ Std) |
|---|---|---|
| Ant-v4 | $1.00 \pm 0.00$ | $8.80 \pm 3.31$ |
| HalfCheetah-v4 | $1.00 \pm 0.00$ | $9.80 \pm 1.94$ |
| Swimmer-v4 | $1.00 \pm 0.00$ | $3.60 \pm 4.72$ |
| Walker2d-v4 | $1.00 \pm 0.00$ | $2.80 \pm 3.12$ |

Table 17: Number of repeated computation loops required in Alg. 1 (Mean $\pm$ Std), using random initializations until the Expertise Dispersion Criterion are satisfied

Table 18: Performance comparison of BC, ILEED, and Scoring (ours), reported as mean ± std of reward. The "best policy" refers to the checkpoint with the highest true reward (oracle) over the entire training process, while "last 5 policies mean" denotes the average performance over the final five training checkpoints.

| Environment | BC (best policy) | ILEED (best policy) | BC (last 5 policies mean) | ILEED (last 5 policies mean) | Scoring (ours), (last 5 policies mean) |
|---|---|---|---|---|---|
| **General expertise test** | | | | | |
| Ant-v4 | 2088.5 ± 42.0 | 2188.6 ± 70.3 | 1341.8 ± 164.5 | 1428.2 ± 80.3 | **6084.6 ± 146.9** |
| HalfCheetah-v4 | 3352.5 ± 214.2 | 3545.7 ± 173.0 | 2318.1 ± 521.2 | 1526.2 ± 1020.4 | **8751.2 ± 1797.9** |
| Swimmer-v4 | 329.4 ± 17.7 | 344.6 ± 1.4 | 21.3 ± 23.0 | 16.8 ± 20.5 | **305.4 ± 15.7** |
| Walker2d-v4 | 1472.7 ± 127.9 | 1439.2 ± 97.2 | 308.4 ± 111.5 | 440.3 ± 313.0 | **3959.2 ± 671.4** |
| **Low expertise test** | | | | | |
| Ant-v4 | 2033.2 ± 55.7 | 2041.3 ± 32.6 | 1473.4 ± 97.7 | 1376.5 ± 62.1 | **5932.0 ± 335.2** |
| HalfCheetah-v4 | 4077.1 ± 24.9 | 4074.9 ± 33.3 | 3753.9 ± 220.5 | 3767.6 ± 176.9 | **7335.0 ± 1316.4** |
| Swimmer-v4 | 220.3 ± 71.3 | 255.2 ± 54.7 | 27.8 ± 11.6 | 34.0 ± 2.4 | **195.4 ± 92.1** |
| Walker2d-v4 | 1345.3 ± 110.0 | 1509.3 ± 213.9 | 436.6 ± 69.1 | 605.3 ± 94.5 | **2961.8 ± 339.8** |

Table 19: Estimated expertise levels $\{\hat{\alpha}_i\}$ for Robomimic data (Can, Lift, and Square tasks) for **Stage 1**. Mean estimates and mean ± std error across seeds shown. True values listed separately.

| Env | Type | 1 | 2 | 3 | 4 | 5 | 6 | Mean error ± std |
|---|---|---|---|---|---|---|---|---|
| **General expertise test [0.1 - 0.9]** | | | | | | | | |
| Can | Est. | 0.3464 | 0.4574 | 0.5639 | 0.6751 | 0.7833 | 0.8908 | 0.123 ± 0.083 |
| | True | 0.1000 | 0.2600 | 0.4200 | 0.5800 | 0.7400 | 0.9000 | - |
| Lift | Est. | 0.2039 | 0.3462 | 0.4837 | 0.6238 | 0.7612 | 0.8978 | 0.054 ± 0.035 |
| | True | 0.1000 | 0.2600 | 0.4200 | 0.5800 | 0.7400 | 0.9000 | - |
| Square | Est. | 0.2802 | 0.3961 | 0.5175 | 0.6299 | 0.7446 | 0.8623 | 0.084 ± 0.060 |
| | True | 0.1000 | 0.2600 | 0.4200 | 0.5800 | 0.7400 | 0.9000 | - |
| **Low expertise test [0.05 - 0.35]** | | | | | | | | |
| Can | Est. | 0.3267 | 0.3678 | 0.4026 | 0.4446 | 0.4818 | 0.5214 | 0.224 ± 0.036 |
| | True | 0.0500 | 0.1100 | 0.1700 | 0.2300 | 0.2900 | 0.3500 | - |
| Lift | Est. | 0.1863 | 0.2403 | 0.2930 | 0.3453 | 0.3946 | 0.4490 | 0.118 ± 0.013 |
| | True | 0.0500 | 0.1100 | 0.1700 | 0.2300 | 0.2900 | 0.3500 | - |
| Square | Est. | 0.2819 | 0.3182 | 0.3688 | 0.4064 | 0.4477 | 0.4942 | 0.186 ± 0.030 |
| | True | 0.0500 | 0.1100 | 0.1700 | 0.2300 | 0.2900 | 0.3500 | - |

Table 20: Classification / Labeling results (mean ± std) for Robomimic data. Stage 1: labeling using the pretrained scoring model. TP (true positive), TN (true negative), FP (false positive), FN (false negative).

| Environment | Stage | TP | TN | FP | FN | Accuracy / Precision |
|---|---|---|---|---|---|---|
| **General expertise test [0.1 - 0.9]** | | | | | | |
| Can | 1 | 14369.00 ± 168.56 | 10784.80 ± 182.42 | 4215.20 ± 182.42 | 631.00 ± 168.56 | 0.8385 / 0.7733 |
| Lift | 1 | 14550.30 ± 224.90 | 11966.80 ± 1193.67 | 3033.20 ± 1193.67 | 449.70 ± 224.90 | 0.8839 / 0.8309 |
| Square | 1 | 14344.00 ± 353.01 | 11904.13 ± 980.05 | 3095.87 ± 980.05 | 656.00 ± 353.01 | 0.8749 / 0.8247 |
| **Low expertise test [0.05 - 0.35]** | | | | | | |
| Can | 1 | 5429.20 ± 202.96 | 16703.80 ± 782.11 | 7296.20 ± 782.11 | 570.80 ± 202.96 | 0.7378 / 0.4280 |
| Lift | 1 | 5590.80 ± 221.80 | 18457.00 ± 1843.70 | 5543.00 ± 1843.70 | 409.20 ± 221.80 | 0.8016 / 0.5156 |
| Square | 1 | 5511.47 ± 215.48 | 18226.80 ± 1568.52 | 5773.20 ± 1568.52 | 488.53 ± 215.48 | 0.7913 / 0.4980 |

Table 21: Training Parameters

| Stage | Parameter | Value |
|---|---|---|
| Stage 1 | $f_\phi$ learning rate | $1 \times 10^{-4}$ |
| | $f_\phi$ batch size | 1000 |
| | $f_\phi$ optimizer | Adam |
| | Variance threshold $\epsilon$ for expertise levels | 0.0005 |
| | Expertise levels estimation frequency | Every 5 epochs |
| | $f_\phi$ training epochs | 50 (Ant, HalfCheetah, Swimmer) |
| | | 100 (Walker2d) |
| Stage 2 | $\pi_\theta$ model | SAC (Stable-Baselines3) |
| | $\pi_\theta$ learning starts from | 100 steps |
| | $\pi_\theta$ batch size | 1024 |
| | $\pi_\theta$ learning rate | $2 \times 10^{-3}$ |
| | $\pi_\theta$ replay buffer size | $1 \times 10^6$ |
| | $\pi_\theta$ training steps / IL iteration | $1.5 \times 10^4$ |
| | $f_\phi$ learning rate | $1 \times 10^{-4}$ |
| | $f_\phi$ batch size | 1024 |
| | $f_\phi$ optimizer | Adam |
| | $f_\phi$ updates epochs / IL iteration | 500 |
| | $\Delta_t, p$ for relabeling early stopping | 30 (for 5 consecutive steps) |
| | IL total iterations | 200 |
| | checkpoints saving freqency | 1 per IL iteration |