# OpenReview forum: "From Many Imperfect to One Trusted: Imitation Learning from Heterogeneous Demonstrators with Unknown Expertise"
_ICLR.cc/2026/Conference — Submitted to ICLR 2026_

### Official Review · Reviewer_Hadp · 2025-10-24

**Soundness:** 3
**Presentation:** 3
**Contribution:** 2
**Rating:** 4
**Confidence:** 3

**Summary:**

The submission studies imitation learning from demonstrators with different levels of expertise, assigning each a scalar expertise value and treating trajectories as mixtures of optimal and suboptimal behavior. It introduces a two-stage pipeline: an EM-style algorithm that learns an optimality score function and the expertise levels, followed by policy training that fine-tunes the policy with stopping, mixup, and an agent-matching penalty. Stage 1 has a convergence guarantee. On MuJoCo tasks across general and very low expertise regimes, the method outperforms GAIL, RIL, and WGAIL, with ablation showing the effectiveness of relabeling.

**Strengths:**

- The problem formulation is clear, and the approach is reasonable.
- A standard convergence analysis of the EM algorithm is provided for the first stage of the proposed method.
The proposed method is tested on MuJoCo and Gymnasium tasks.
- An ablation study on relabeling, early stopping, and top-k selection.
- Two sets of experiments—the “general expertise test” and “low expertise test”—demonstrate the effectiveness of the proposed method.

**Weaknesses:**

- The model for the expertise level is too simple compared to (https://arxiv.org/pdf/2202.01288) mentioned by the paper. The current formulation may miss demonstrators who are experts in some regions and poor in others.
- The criteria (line 253 and line 264) discussed after Theorem 1 make the approach ad hoc and require high quality supervision (albeit a small amount).
- ILEED is missing from the experiment.

**Questions:**

- What is the usage of alpha-hat_i in Algorithm 2?
- Why is it fine to choose alpha’ = 0.5 (line 209)? What could a misspecified outcome be? How can the misspecified case be handled?

---

> ### Author Response · Authors · 2025-11-23
> **Thank you for your constructive comments**
>
> > **W1: Modeling of expertise level is simple. The current formulation may miss demonstrators who are experts in some regions and poor in others.**
>
> A: Thank you for the comments. The expertise level $\alpha_i$ in our paper is just an intermediate variable to describe the general skill ability of expert $i$. We also have a **state-action dependent expertise level** in our paper, which is actually the **optimality score** that is defined as the probability that (s, a) is optimal given its values: P(z = 1|s, a). This can be viewed as a sample-specific, conditional expertise level. We hope this clarifies that our approach captures both general and context-dependent expertise.

---

> > ### Author Response · Authors · 2025-11-23
> >
> > > **W2.1: Expertise dispersion criterion makes the approach ad-hoc.**
> >
> > A: Thank you for the comments. Our theoretical guarantees for Algorithm 1 follow classical EM results [1] and recent analyses of alternating/minimax procedures in imitation learning [2]: the algorithm is guaranteed to converge to **stationary points**. As is standard for EM-type methods, the quality of the solution depends on initialization. Thus, a principled initialization strategy is necessary.
> >
> > To address this, we introduce the *expertise-dispersion criterion* to select good initializations and avoid degenerate stationary points. We run the algorithm with multiple random initializations until the criterion is met. Importantly, our experiments show that even in extremely low-expertise regimes (see Table 2), the stationary points reached by our method remain highly effective in practice.
> >
> > We further added an ablation study on the **number of initializations (Table 17 in the updated pdf)**. For General Expertise settings, a *single initialization* suffices; in more challenging Low Expertise cases, the number increases but remains **below 10**.
> >
> > | Env             | General Expertise (Mean ± Std) | Low Expertise (Mean ± Std) |
> > |-----------------|-------------------------------|----------------------------|
> > | Ant-v4          | 1.00 ± 0.00                    | 8.80 ± 3.31               |
> > | HalfCheetah-v4  | 1.00 ± 0.00                    | 9.80 ± 1.94               |
> > | Swimmer-v4      | 1.00 ± 0.00                    | 3.60 ± 4.72               |
> > | Walker2d-v4     | 1.00 ± 0.00                    | 2.80 ± 3.12               |
> >
> > [1] On the convergence properties of the EM algorithm, The Annals of statistics, 1983
> >
> > [2] On Computation and Generalization of Generative Adversarial Imitation Learning, ICLR 2020

---

> ### Author Response · Authors · 2025-11-23
>
> > **W2.2: Optimality alignment criterion requires high-quality supervision.**
>
> A: Thank you for the comment. It is true that our method requires a **small amount of supervision**, since we have **no access to labels for sample optimality** or **expertise levels**. By the **No Free Lunch** theorem, without structural assumptions or any supervision, no learning algorithm can reliably recover correct labels.
>
> In such an unsupervised setting, our classification method effectively performs an operation similar to clustering on these unlabeled data. A limited number of human queries is then used to ensure that the resulting clusters are assigned the correct labels. Importantly, the expert only needs to indicate whether the **majority of predicted labels are correct**, rather than verifying each sample individually. We hope this clarifies why our method uses minimal supervision while remaining effective.
>
> We have added an ablation study on the **number of human queries** (see Figure 6 and Tables 13–14 in the revised PDF). Below we summarize the results.
>
> **Table: Classification Results of the Scoring Model on Imperfect Demonstrations (Ant-v4)**
>
> **Metrics:**  Accuracy / Precision (%)
>
> **Note:** Values in the *Improvement* columns represent Stage 1 → Stage 2 relative improvement (abbreviated as Impr. in the table) (%).
>
> *Ablation study on expert query count.*
>
> | **Test**    | **Stage** | **Query 1** | **Impr.** | **Query 3** | **Impr.** | **Query 5** | **Impr.** | **Query 7** | **Impr.** | **Query 9** | **Impr.** |
> |-------------|-----------|-------------|-----------|-------------|-----------|-------------|-----------|-------------|-----------|-------------|-----------|
> | **General** | 1 | 99.93 / 99.97 | – | 99.92 / 99.94 | – | 99.91 / 99.89 | – | 99.92 / 99.96 | – | 99.94 / 99.95 | – |
> | **Expertise** | 2 | 99.92 / 99.92 | −0.01 / −0.05 | 99.90 / 99.90 | −0.02 / −0.04 | 99.89 / 99.89 | −0.02 / 0.00 | 99.89 / 99.89 | −0.03 / −0.07 | 99.91 / 99.91 | −0.03 / −0.04 |
> | **Low** | 1 | 96.26 / 75.31 | – | 97.33 / 80.17 | – | 90.33 / 51.78 | – | 97.02 / 78.81 | – | 97.14 / 79.26 | – |
> | **Expertise** | 2 | 99.64 / 98.22 | **+3.51 / +30.42** | 99.52 / 97.60 | **+2.25 / +21.74** | 99.65 / 98.23 | **+10.32 / +89.71** | 99.48 / 97.40 | **+2.54 / +23.59** | 99.73 / 98.65 | **+2.67 / +24.46** |
>
>
> **Table: Ablation study on Query times for Ant-v4.**
>
> Reported as mean ± std of Reward  across random seeds; each seed's value is the mean over its last five checkpoints.
>
> | Query times | Expertise [0.1–0.9]   | Expertise [0.05–0.15] |
> |------------|----------------------|-----------------------|
> | 1          | 5433.6 ± 365.6       | 5810.2 ± 301.3        |
> | 3          | 5655.2 ± 307.3       | 6011.2 ± 219.9        |
> | 5          | 6084.6 ± 146.9       | 5932.0 ± 335.2        |
> | 7          | 5981.4 ± 264.6       | 5944.2 ± 279.8        |
> | 9          | 5881.6 ± 265.3       | 6130.8 ± 126.8        |
>
> **Summary:**
> We observe that even a **single query** yields performance comparable to using 5 or more queries. This is because Stage-1 classification accuracy is already extremely high (>99%), making incorrect cluster assignment highly unlikely. Likewise, the learned policies are robust across different query counts.
> We hope this clarifies that our method uses **minimal supervision** while remaining both effective and practical.

---

> > ### Author Response · Authors · 2025-11-23
> >
> > > **W3: Experimental comparison with ILEED.**
> >
> > A: Thanks for the suggestion. In the first draft, we didn’t compare with ILEED because it is an offline method, and all other baselines are online methods. We now **added experimental comparisons with ILEED**. As shown in the table below and Table 2, Figure 3 in the updated pdf, our methods achieve performance improvements of at least 90% over ILEED.
> >
> > *Performance comparison of ILEED, and Scoring (ours), reported as mean $\pm$ std of reward.*
> >
> > | Environment      | ILEED     | Scoring (ours)  |
> > |------------------|--------------------------|-------------------------------|
> > | **General expertise test** |                          |                               |
> > | Ant-v4           | 1428.2 ± 80.3            | **6084.6 ± 146.9**            |
> > | HalfCheetah-v4   | 1526.2 ± 1020.4          | **8751.2 ± 1797.9**           |
> > | Swimmer-v4       | 16.8 ± 20.5              | **305.4 ± 15.7**              |
> > | Walker2d-v4      | 440.3 ± 313.0            | **3959.2 ± 671.4**            |
> > | **Low expertise test** |                    |                               |
> > | Ant-v4           | 1376.5 ± 62.1            | **5932.0 ± 335.2**            |
> > | HalfCheetah-v4   | 3767.6 ± 176.9           | **7335.0 ± 1316.4**           |
> > | Swimmer-v4       | 34.0 ± 2.4               | **195.4 ± 92.1**              |
> > | Walker2d-v4      | 605.3 ± 94.5             | **2961.8 ± 339.8**            |
> >
> >
> >
> > Compared to our method, **ILEED** is purely **offline** and relies on maximum-likelihood estimation~(MLE), making it sensitive to **distribution shift** when the learned policy visits new states not present in demonstrations. As shown in the table below (Table 18 in updated pdf),  the reward of ILEED averaged over the last five policies (where the policy loss function attempts to maximize likelihood) is much lower than the reward obtained by their best policy encountered during training (which serves as an oracle). This may be due to the distribution shift or unstable training.
> >
> > *The “best policy” refers to the checkpoint with the highest true reward (oracle) over the entire training process, while “last 5 policies mean” denotes the average performance over the final five training checkpoints.*
> >
> > | Environment       | ILEED (best policy)      | ILEED (last 5 policies mean) | Scoring (ours), (last 5 policies mean) |
> > |-------------------|---------------------------|-------------------------------|-----------------------------------------|
> > | **General expertise test** |                           |                               |                                         |
> > | Ant-v4            | 2188.6 ± 70.3             | 1428.2 ± 80.3                 | 6084.6 ± 146.9                      |
> > | HalfCheetah-v4    | 3545.7 ± 173.0            | 1526.2 ± 1020.4               | 8751.2 ± 1797.9                     |
> > | Swimmer-v4        | 344.6 ± 1.4               | 16.8 ± 20.5                   | 305.4 ± 15.7                        |
> > | Walker2d-v4       | 1439.2 ± 97.2             | 440.3 ± 313.0                 | 3959.2 ± 671.4                      |
> > | **Low expertise test** |                       |                               |                                         |
> > | Ant-v4            | 2041.3 ± 32.6             | 1376.5 ± 62.1                 | 5932.0 ± 335.2                      |
> > | HalfCheetah-v4    | 4074.9 ± 33.3             | 3767.6 ± 176.9                | 7335.0 ± 1316.4                     |
> > | Swimmer-v4        | 255.2 ± 54.7              | 34.0 ± 2.4                    | 195.4 ± 92.1                        |
> > | Walker2d-v4       | 1509.3 ± 213.9            | 605.3 ± 94.5                  | 2961.8 ± 339.8                      |
> >
> >
> > In comparison, **our method** first learns a scoring model offline, then uses it as a reward and **iteratively refines** it using on-policy rollouts with mixup, agent-matching, and relabeling, so to explicitly addresses covariate shift, a well-known weakness of offline IL, and produces significantly more robust policies.
> >
> > We hope these results address the reviewer’s concerns.

---

> ### Author Response · Authors · 2025-11-23
>
> > **Q1: alpha-hat_i in Algorithm 2.**
>
> A: Thanks for pointing out the confusion,
> $\hat\alpha_i$  refers to the estimated expertise levels, and $\alpha_i$ refers to the true value of expertise levels. For better consistency. We have updated the symbol used in Alg 1 from alpha_i to  alpha-hat_i.
>
> > **Q2: Why choose $\alpha’ = 0.5$?**
>
> A: Choosing $\alpha' = 0.5$ corresponds to using a balanced version of the error rate, which is widely chosen as an appropriate performance measure when one class dominates the other; in such cases, minimizing the standard classification error often leads to poor performance on the minority class [1, 2, 3].
>
> In our setting, $\alpha'$ reflects the expertise level of the agent, which evolves from novice to expert during training. The labels $z=1$ and $z=0$ can become highly imbalanced over the course of training. Therefore, we set  $\alpha' = 0.5$ to assign equal weight to false positives and false negatives, thereby preventing the scoring model from sacrificing accuracy on the minority class in favor of the dominant class.
>
> [1] The balanced accuracy and its posterior distribution, ICPR 2010
>
> [2] Learning from corrupted binary labels via class-probability estimation, ICML 2015
>
> [3] On Symmetric Losses for Learning from Corrupted Labels, ICML 2019
>
> To support this choice, we have added ablation studies examining the effect of different values of $\alpha'$. Table 1 shows the classification results of the scoring model and Table 2 shows the reward obtained by the learned policies. The results support that the selection of  $\alpha' = 0.5$ can generate best results for both General expertise and Low expertise tests.
>
> **Table 1: Classification results of the scoring model on imperfect demonstrations for Stage 1.**
>
> Accuracy and Precision (%, mean ± std), for Ant-v4 across different $\alpha’$ values (expected expertise in the target environment). Supplementary results see Fig. 4.
>
> | $\alpha’$ | Accuracy (General) | Precision (General) | Accuracy (Low) | Precision (Low) |
> |-------------|------------------|-------------------|----------------|----------------|
> | 0.1         | 59.97 ± 19.94    | 20.00 ± 39.99     | 90.00 ± 0.00   | 0.00 ± 0.00    |
> | 0.2         | 89.94 ± 19.97    | 79.97 ± 39.99     | 90.00 ± 0.00   | 0.00 ± 0.00    |
> | 0.3         | 99.92 ± 0.02     | 99.96 ± 0.03      | -              | -              |
> | 0.4         | 99.94 ± 0.01     | 99.98 ± 0.02      | 90.00 ± 0.00   | 0.00 ± 0.00    |
> | 0.45        | -                | -                 | 96.44 ± 2.43   | 88.46 ± 11.92  |
> | 0.5         | 99.91 ± 0.03     | 99.89 ± 0.07      | 90.33 ± 2.45   | 51.78 ± 7.56   |
> | 0.55        | -                | -                 | 81.07 ± 35.54  | 77.18 ± 34.12  |
> | 0.6         | 99.92 ± 0.02     | 99.90 ± 0.05      | 20.03 ± 20.07  | 12.01 ± 4.02   |
> | 0.7         | 99.93 ± 0.02     | 99.93 ± 0.05      | 27.21 ± 34.42  | 22.35 ± 12.70  |
> | 0.8         | 99.92 ± 0.02     | 99.90 ± 0.04      | -              | -              |
> | 0.9         | 79.91 ± 24.43    | 79.87 ± 24.39     | 10.00 ± 0.00   | 10.00 ± 0.00   |
>
> ---
>
> **Table 2: Ablation study on $\alpha’$ for Ant-v4.**
>
> Reported as mean ± std of reward across random seeds; each seed's value is the mean over its last five checkpoints. Note: no policy learning results for expertise [0.05–0.15] when $\alpha’ \le 0.4$ because all imperfect demonstrations are labeled sub-optimal, leaving the optimal set empty.
>
> | $\alpha’$ | Expertise [0.1–0.9] | Expertise [0.05–0.15] |
> |-----------------|--------------------|-----------------------|
> | 0.1             | 1149.4 ± 2298.8    | --                    |
> | 0.2             | 3700.6 ± 2286.5    | --                    |
> | 0.3             | 5241.6 ± 957.9     | --                    |
> | 0.4             | 5308.0 ± 377.7     | --                    |
> | 0.45            | --                 | 5509.0 ± 493.4       |
> | 0.5             | 6084.6 ± 146.9     | 5932.0 ± 335.2       |
> | 0.55            | --                 | 5496.6 ± 509.1       |
> | 0.6             | 5579.0 ± 416.2     | 2927.8 ± 2064.2      |
> | 0.7             | 5542.8 ± 261.6     | 3497.6 ± 2349.8      |
> | 0.8             | 5262.6 ± 330.8     | --                    |
> | 0.9             | 5090.2 ± 573.1     | 3171.0 ± 1910.6      |

---

> ### Author Response · Authors · 2025-11-27
> **Kind Reminder**
>
> Dear Reviewer Hadp,
>
> Since we have provided **additional theoretical** and **experimental clarifications** addressing the concerns you raised, particularly regarding the **comparisons with ILEED**, and given that you have acknowledged our contributions in your previous reply, we kindly ask whether you could raise your score to reflect these clarifications?
>
> Best regards,
> Authors of Submission 20336 (From Many Imperfect to One Trusted: Imitation Learning from Heterogeneous Demonstrators with Unknown Expertise)

---

### Official Review · Reviewer_yeCu · 2025-10-29

**Soundness:** 2
**Presentation:** 2
**Contribution:** 2
**Rating:** 4
**Confidence:** 4

**Summary:**

The paper tackles imitation learning (IL) when demonstrations come from heterogeneous demonstrators of unknown and widely varying expertise. The authors model each demonstrator as a mixture policy under a latent optimality label on state–action pairs. Stage 1 learns (i) a state–action optimality scorer via surrogate demonstrator classification, and (ii) demonstrator expertise levels by an EM‑style alternating procedure. Stage 2 uses a surrogate reward to train a policy with SAC, while iteratively refining using on‑policy rollouts, a negative (agent‑matching) term, and a mixup regularizer, plus a top‑k pseudo‑labeling scheme with an early‑stopping heuristic. On MuJoCo control tasks, the method outperforms GAIL, RIL, WGAIL and approaches performance of an oracle GAIL trained on hand‑selected optimal subsets, including regimes where all demonstrators are highly suboptimal.

**Strengths:**

- The paper addresses a realistic data regime, i.e., many imperfect demonstrators with unknown quality, and proposes a concrete way to mine signal without explicit optimality labels.
- Leveraging surrogate set classification (SSC) to recover P(z=1∣x) from multi‑set membership is a smart reduction that connects demonstrator‑ID prediction to optimality scoring with a known transform.
- The refinement loop (on‑policy rollouts, an agent‑matching penalty, and mixup) directly targets covariate shift and over‑confidence issues that often hurt IL, and the ablations help isolate these effects.

**Weaknesses:**

- The EM‑style analysis is potentially incorrect in places. In Theorem 1, the “M‑step” is written with a minimization over $\(\phi, \alpha\)$ that still plugs in $\phi_t$, and the “E‑step” uses a hard 0.5 threshold instead of the expected posterior E[z∣s,a]. More importantly, identifiability of the expertise priors $\{\alpha_i\}$ and the optimality scorer via SSC requires strong conditions (e.g., class‑conditional distributions invariant across sets and “mutual irreducibility” assumptions); these are not clearly stated in the main text, yet they are central to SSC’s guarantees. Without them, $\alpha$ and $f_\phi$ can be non‑identifiable or flip‑ambiguous. It it necessary to make the assumptions explicit and align the proof to standard EM or variational lower‑bound updates.
- Eq. (1) implicitly assumes a single shared suboptimal policy for all demonstrators. In real data, different novices commit different types of errors and visit different states. SSC’s reduction typically assumes shared class‑conditionals p(x∣z) across sets; the paper’s occupancy‑measure formulation includes demonstrator‑dependent state visitation, which can violate these assumptions and bias the recovered scorer. What if each demonstrator has a different structured suboptimality with domain shift?
- Here are places where the exposition is inconsistent or confusing: the top‑k selection text conflicts with the $f_\phi(s, a) > 0.5$ rule (lowest vs. highest scores), Algorithm 1’s steps don’t align with the proof, and Theorem 1’s objective/updates are misstated.
- Closely related works are not cited or are under‑discussed:
  - AIRL for learning disentangled, portable rewards from demos [1].
  - T‑REX/D‑REX for better‑than‑demonstrator performance from suboptimal data via ranking [2–3].
  - VPIL for vague feedback over demos [4].

## References

[1] Learning Robust Rewards with Adversarial Inverse Reinforcement Learning.

[2] Extrapolating Beyond Suboptimal Demonstrations via Inverse Reinforcement Learning from Observations.

[3] Better-than-Demonstrator Imitation Learning via Automatically-Ranked Demonstrations.

[4] Imitation Learning from Vague Feedback.

**Questions:**

Please refer to the weaknesses.

---

> ### Author Response · Authors · 2025-11-23
> **Thank you for your constructive comments**
>
> >**Q1: EM-style analysis is incorrect in places: M-step plugs in $\phi_t$ and E step uses a hard 0.5 threshold. Identifiability of expertise priors and optimality scorer requires class‑conditional distributions invariant across sets and “mutual irreducibility” assumptions. Flip‑ambiguous issues.**
>
> A: Thank you for pointing out these concerns.
>
> **EM-style analysis**
>
> First, we clarify our EM algorithm 1. We start with initializing the scoring model $f_\phi$ and the expertise levels $\{ \alpha_i \}_{i=1}^m$.
>
> - **In the M-step**, we **fix the expertise levels** and update the scoring model \(f_\phi\) by minimizing the surrogate demonstrator classification (SSC) risk (Eq. (4)). This explains why \(\phi_t\) appears in the M-step, where \(t\) denotes the current iteration.
>
> - **In the E-step**, we **fix the scoring model** and estimate the expertise levels.
>
> The convergence guarantee for this EM-style algorithm follows the **standard EM convergence theory** [6].
>
> We use a **0.5 threshold** because the SSC training is proven to recover the Bayes optimal classifier and is an accurate class probability estimator, given a proper loss is used and the model class is well-specified in the sense that the true Bayes optimal is included [1].
>
> To make this explicit, we added **Lemma 1** in the updated manuscript to reflect this point. In learning theory, it is known that the Bayes optimal classifier is thresholded at $0.5$ to output the optimal decision rule under $0$-$1$ loss [2,3]. This explains the choice of threshold $0.5$ for predicting the optimality labels, which is consistent with [1].
>
> **Identifiability**
>
> It is true that identifiability of the optimality scorer requires **invariant class-conditional distributions** when the expertise priors are known, as shown in [1]. When the priors are *not* known, the **mutual irreducibility assumption** [4] is needed for identifiability of the expertise priors.
>
> To clarify this point, we added Footnote 4 in Section 3.2.2:
>
> > *“To make the expertise-level estimation problem solvable, we assume the existence of anchor points in the demonstration dataset, i.e., data points that belong to class \(z=1\) and \(z=0\) almost surely.”*
>
> Our assumed anchor points are proved to be equivalent to the mutually irreducible condition in [5]; please refer to their Section 6.1 and Appendix H for the proof.
>
> **Flip-ambiguous issues**
>
> Our theoretical guarantees follow classical EM results [6] and recent analyses of alternating/minimax procedures in imitation learning [7]: algorithm 1 is guaranteed to converge to **stationary points**, not necessarily global optima. Consequently, flip-ambiguity is theoretically unavoidable.
>
> To mitigate this, we incorporate mechanisms such as the *expertise-dispersion criterion* to select good initializations, and *optimality alignment criterion* for preventing label flips, helping the algorithm avoid degenerate stationary points. Our experiments further demonstrate that, even in the extremely low-expertise regime (see Table 2), the stationary points reached by our algorithm are practically effective.
>
> ---
>
> **References**
>
> [1] Binary classification from multiple unlabeled datasets via surrogate set classification, ICML 2021
>
> [2] Convexity, Classification, and Risk Bounds, Journal of the American Statistical Association 2006
>
> [3] Understanding machine learning: From theory to algorithms, Cambridge university press, 2014
>
> [4] Mixture Proportion Estimation via Kernel Embeddings of Distributions, ICML 2016
>
> [5] Learning from Corrupted Binary Labels via Class-Probability Estimation, ICML 2015
>
> [6] On the convergence properties of the EM algorithm, The Annals of statistics, 1983
>
> [7] On Computation and Generalization of Generative Adversarial Imitation Learning, ICLR 2020

---

> > ### Author Response · Authors · 2025-11-23
> >
> > > **Q2.1: The occupancy‑measure formulation includes demonstrator‑dependent state visitation, which can violate the SSC assumptions.**
> >
> > A: We thank the reviewer for raising this subtle and important point!
> >
> > The concern is valid: if the occupancy model assumed that each demonstrator has their own RL-induced state visitation distribution $\rho^{\pi_i}(s)$, then the resulting joint distribution $\rho_i(s,a)$ would no longer be a mixture of *shared* optimal and suboptimal components, violating the assumption of SSC. In such a case, different demonstrators would exhibit covariate shift between their optimal and suboptimal components, which would indeed require additional correction techniques (e.g., density-ratio-based reweighting) to recover identifiability of the optimal and suboptimal occupancy components.
> >
> > However, we clarify that our paper does *not* assume that $\rho^{\pi_i}(s)$ is the RL visitation distribution induced by executing the mixed policy $\pi_i$. Instead, it is induced by the occupancy measure of demonstrator-$i$ via $\rho_i(s)=\sum_a \rho_i(s,a)=\alpha_i \rho^{\pi^*}(s)+(1-\alpha_i)\rho^{\pi^{\mathrm{sub}}}(s)$, which in general differs across demonstrators and is not equal to the RL visitation distribution. Therefore it does not violate the SSC assumptions.
> >
> > We have added a footnote in Section 3.1 clarifying this point to avoid confusion.
> >
> > > **Q2.2: Eq. (1) assumes a single shared suboptimal policy for all demonstrators. What if each demonstrator has a different structured suboptimality?**
> >
> > A: Thank you for the interesting question. If each demonstrator were allowed to have an **arbitrary and fully flexible** suboptimal policy, with no structure linking these policies, then the optimal component would become **unidentifiable**. In that case, the mixture in Eq. (1) can be explained in **infinitely many** ways, making the problem statistically **unsolvable**.
> >
> > That said, it is indeed possible to extend our framework to allow demonstrator-specific suboptimality under **structured assumptions**. For example, one could let
> > $$\rho_i^{\mathrm{sub}}(s,a)=\rho_i^{\mathrm{sub}}(s)\,\pi^{\mathrm{sub}}(a\mid s),$$
> > while keeping the conditional action distribution \(\pi^{\mathrm{sub}}(a\mid s)\) shared across demonstrators.
> > Differences between demonstrators would then be captured by a density ratio
> > $$\rho_i^{\mathrm{sub}}(s)=r_i(s)\,\rho^{\pi^{\mathrm{sub}}}(s),\quad r_i(s)>0,\quad \mathbb{E}[r_i(s)]=1.$$
> > Such a structure would allow one to normalize each demonstrator’s suboptimal occupancy via importance weighting,
> > $$\tilde{\rho}_i^{\mathrm{sub}}(s,a) = \frac{1}{r_i(s)}\,\rho_i^{\mathrm{sub}}(s,a),$$
> > thereby recovering a shared suboptimal component.
> >
> > We believe that analyzing structured suboptimality extensions is a promising direction, but it is beyond the scope of the current paper. We will definitely explore it in future work!

---

> ### Author Response · Authors · 2025-11-23
>
> > **Q3: Confusing exposition.**
>
> > **Q3.1 Top-k selection text conflicts with the $f_\phi(s,a) > 0.5$ condition**
>
> A: Thank you for pointing out this confusion. We clarify the rationale behind using $f_\phi(s,a) > 0.5$ in Stage 1 and switching to top-k selection in Stage 2.
>
> Before Stage 2 (Algorithm 2), the policy has not yet been improved, so $f_\phi(s,a) > 0.5$ is a natural and fair criterion for distinguishing optimal vs. non-optimal samples under the initial classifier.
> However, during Stage 2, as the agent $\pi_\theta$ becomes stronger, $f_\phi$ becomes increasingly “confused” by the agent’s higher-quality rollouts; their dominance creates a highly imbalanced learning scenario.
>
> Therefore, instead of continuing to use the fixed threshold $f_\phi(s,a) > 0.5$, we switch to a top-k selection strategy in Stage 2 to focus on the relatively most confident samples as training progresses.
>
> We also added an ablation study on the choice of the top-k fraction. Results are shown below (also see Figure 2 and Table 10 in the updated pdf). The results show that more pessimistic (smaller $k$) values generally lead to better performance.
>
> **Table: Ablation study on top-k hyperparameter on Ant-v4.**
> Values are mean ± std across random seeds (each seed averaged over its last five checkpoints).
>
> | Top-k fraction | Expertise [0.1–0.9]          | Expertise [0.05–0.15]        |
> |----------------|-------------------------------|-------------------------------|
> | 0.05           | --                            | 5887.6 ± 263.8                |
> | 0.1            | 5403.8 ± 378.7                | **5932.0 ± 335.2**            |
> | 0.15           | --                            | 3663.4 ± 1950.9               |
> | 0.2            | --                            | 1756.6 ± 902.6                |
> | 0.3            | 5809.0 ± 274.8                | --                            |
> | 0.5            | **6084.6 ± 146.9**            | --                            |
> | 0.55           | 4109.2 ± 2204.9               | --                            |
> | 0.6            | 3641.0 ± 1237.7               | --                            |
> | 0.7            | 2383.0 ± 1059.7               | --                            |
>
>
> > **Q3.2 Algorithm 1’s steps don’t align with the proof, and Theorem 1’s objective/updates are misstated**
>
> A: Thanks for the comments. We clarify the consistency between Algorithm 1’s steps and Theorem 3 (previously Theorem 1).
>
> For E step, in both Algorithm 1 and Theorem 3 (previously Theorem 1), the updating equations are all same:
> $\alpha_i \leftarrow \frac{ \sum_{(s,a) \in D_i} \left[ f_\phi(s,a) > 0.5 \right] }{||D_i||},~\forall i\in[m]$
>
> For M step, the loss function is computed with estimated expertise level $\alpha$ from E step. Then the objective function does not include $\alpha$ as a variable (see Eq. 4 in updated pdf).
>
> However, in the proof of Theorem 3 (previously Theorem 1) or from the perspective of whole EM-style algorithm, both $\alpha$  and $\theta$ are variables in the objective function, thus, the updates and objective function is expressed as Eq. 5  in updated pdf.
>
>
> We hope our clarification can address the confusion.

---

> ### Author Response · Authors · 2025-11-23
>
> > **Q4: Discussions about the recommended papers.**
>
> A: Thanks for pointing out the related papers! We discuss them below.
>
>
> **Comparisons with AIRL [1]**
> Adversarial Inverse Reinforcement Learning (AIRL) extends the GAIL framework by structuring the discriminator to decompose into a state-only reward term and a shaping term, enabling recovery of a reward function that is disentangled from environment dynamics. This structure allows AIRL to learn portable rewards that can be re-optimized in new or modified environments.
>
> Our problem setting, however, is fundamentally different. Rather than assuming demonstrations come from a single (near-optimal) expert, we address imitation learning from heterogeneous, unlabeled, and potentially highly suboptimal demonstrators with unknown expertise levels. Although our method learns a scoring model that serves as a surrogate reward, the objective is not reward recovery or transfer, but robust policy learning when expert dominance does not hold.
> AIRL, like GAIL, is not designed to handle heterogeneous demonstrators and presumes access to expert demonstrations.
> Therefore, their assumptions and objectives cannot address the key challenges in our setting.
>
> ---
>
> **Comparisons with T-REX and D-REX [2-3]**
>
> Both T-REX and D-REX learn a reward function from ranked demonstrations, allowing an agent to potentially outperform the demonstrator via extrapolation. T-REX relies on experts to provide ranking comparisons, while D-REX can automatically generate rankings by injecting noise into a cloned policy, based on the assumption that added noise leads to a continuous degradation in performance.
>
> Compared to T-REX, our method does not require large amounts of supervision or expert ranking feedback during training. Moreover, while D-REX assumes homogeneous demonstrations and introduces heterogeneity artificially, our approach naturally handles heterogeneous datasets and is capable of inferring varying expertise levels within the data. Unlike D-REX, our method does not rely on the assumption that policy performance monotonically decreases with noise injection, which may not hold in more complex domains. In addition, both T-REX and D-REX first learn a surrogate reward function and then optimize a policy using reinforcement learning on this fixed reward, which can lead to covariate shift when the agent encounters states not present in the demonstrations. Our method addresses this issue by continuously fine-tuning the scoring model with new data collected during RL, resulting in improved robustness and generalization. Overall, our approach provides a flexible and practical framework for leveraging demonstrations, particularly in settings with heterogeneous data and limited supervision.
>
> ---
>
> **Comparisons with VPIL [4]**
> Vaguely Pairwise Imitation Learning (VPIL) investigates a weak-supervision setting in which the annotator provides vague pairwise feedback: a pair of demonstrations is labeled only when their quality differs substantially, resulting in two contaminated datasets that still contain mixtures of expert and non-expert trajectories.
> VPIL leverages this pairwise structure to recover the expert occupancy measure through mixture-proportion estimation (MPE) \citep{ramaswamy2016mixture} and risk rewriting, and its COMPILER-E algorithms integrate the recovered distribution into a GAIL-style adversarial imitation framework.
>
> Our problem setting is more general and arguably more challenging.
> Unlike VPIL, which assumes access to vague pairwise comparison labels, we consider a scenario in which the agent receives only heterogeneous demonstrations of unknown and varying quality, with no preference, ranking, or optimality labels.
> In the absence of such supervision, the learner must infer both the underlying demonstration quality and the reward surrogate directly from unlabeled trajectories.
>
> The methodological differences therefore extend beyond supervision assumptions.
> Rather than reconstructing an expert occupancy measure from pairwise datasets, we learn a fine-grained state–action optimality scoring model through a surrogate demonstrator classification objective and an EM-style latent expertise level estimation procedure.
> The trained scoring model then serves as a surrogate reward within an iterative policy-learning loop, enabling robust imitation learning directly from heterogeneous, unlabeled demonstrators without relying on any preference-based supervision.
>
> ---
>
> We have added the above discussions in Appendix A: Further Discussion on Related Work of the updated manuscript.

---

> ### Author Response · Authors · 2025-11-27
> **Kind Reminder**
>
> Dear Reviewer Reviewer yeCu,
>
> Thank you again for your thoughtful comments on the problem formulation and EM analysis. We have revised the paper accordingly, **added new theoretical results** and **experiments** to address your concerns, and cited and discussed the related papers you recommended in a new section. We kindly ask whether you could consider raising your score to reflect these clarifications?
>
> Best regards,
> Authors of Submission 20336 (From Many Imperfect to One Trusted: Imitation Learning from Heterogeneous Demonstrators with Unknown Expertise)

---

### Official Review · Reviewer_JbYb · 2025-10-30

**Soundness:** 3
**Presentation:** 3
**Contribution:** 2
**Rating:** 2
**Confidence:** 4

**Summary:**

This paper considers the imitation learning problem from heterogeneous demonstrators of unknown expertise, proposing a two-stage EM-style framework that jointly estimates demonstrator expertise and learns a state–action optimality scoring model, which is then used as a for policy optimization. The paper is an extension of ILEED (Beliaev et al., ICML 2022) which considered unsupervised expertise estimation for heterogeneous demonstrators. In the authors' formulation, instead of modeling state-dependent embeddings, it models demonstrator expertise as a global scalar mixture coefficient to have a more structured suboptimality.

**Strengths:**

- The problem is important as heterogeneous and imperfect demonstrations are the norm in large-scale IL.
- The paper presents a clean EM formulation, with a clear separation between expertise estimation and policy learning.

**Weaknesses:**

- The novelty is quite incremental relative to ILEED. The new formulation simplifies expertise modeling from state-dependent embeddings to demonstrator-level mixture coefficients and reframes the joint estimation as a classification-based EM procedure.
- The paper should have direct numerical comparison with ILEED as well as other baselines specifically designed for suboptimal demonstrations. The current basedlines mostly cover standard IL methods.
- Evaluations are conducted on synthetic MuJoCo environments where suboptimality is simulated by mixing optimal and degraded SAC policies. The paper will benefit from real human demonstrations like the robomimic dataset.
- The theoretical contribution (EM convergence) is modest and to my best knowledge a well-known proof.

**Questions:**

Can you provide quantitative comparison with ILEED and stronger evidence of scalability or generalization using real human demonstrations?

---

> ### Author Response · Authors · 2025-11-23
> **Thank you for your constructive comments**
>
> >**W1.1: The novelty is quite incremental relative to ILEED.**
>
> A: We thank the reviewer for the comment. However, we would like to clarify that our method is **not incremental to ILEED** and includes **key distinctions and novel contributions**. We provide a comparison between our work and ILEED below.
>
> ---
>
> 1. **Problem setting: uniform noise vs. structured suboptimality**
>
> - **ILEED** considers each demonstrator’s policy as a mixture of the optimal policy and a **uniformly random policy**, assigning equal probability to each action.
>
> - **Our method** removes these restrictions and allows **non-uniform and structured** suboptimal policies.
>
> - **Contribution:** This generalization is crucial for real-world IL, where suboptimal behaviors are *structured* (e.g., consistent skill-specific weaknesses). Our new experiments below show this flexibility significantly improves robustness.
>
> ---
>
> 2. **Expertise modeling: embedding-based vs. optimality-scoring**
>
> - **ILEED** models expertise as a state-dependent sigmoid over state and demonstrator embeddings. Therefore its performance depends heavily on the quality of the recovered embeddings and **assumes that all states are visited by all demonstrators** (since ILEED is purely offline). Sparse or uneven state coverage for a demonstrator can bias expertise estimates, leading to significant policy degradation.
>
> - **Our method** estimates expertise via a **class-prior** $\mathbb{P}(z{=}1)$ and learns a **surrogate scoring model** that directly distinguishes optimal from suboptimal behavior across demonstrators. The model also outputs **state-action dependent expertise** \(\mathbb{P}(z{=}1 \mid s,a)\), called the **optimality score**, which serves as a reward signal for RL.
>
> - **Contribution:** Unlike ILEED, our method **does not require full state-space coverage for each demonstrator**, yielding more robust learning. It also produces a **reward function**, enabling fine-tuning of the scoring model using new samples visited during RL.
>
> ---
>
> 3. **Methodology: offline MLE vs. online refinement**
> - **ILEED** is purely **offline** and relies on maximum-likelihood estimation~(MLE), making it sensitive to **distribution shift** when the learned policy visits new states not present in demonstrations.
>
> - **Our method** first learns a scoring model offline, then uses it as a reward and **iteratively refines** it using on-policy rollouts with mixup, agent-matching, and relabeling.
> - **Contribution:** Our method explicitly addresses covariate shift, a well-known weakness of offline IL, and produces significantly more robust policies.
>
> ---
>
> 4. **Optimization strategy: joint MLE vs. EM + RL**
> - **ILEED** performs joint optimization over policy, state embeddings, and demonstrator embeddings in a single likelihood objective. This tight coupling can lead to instability and degenerate solutions.
>
> - **Our method** uses an EM-style alternating update for estimating expertise and training the scoring model, with a convergence guarantee, followed by standard RL.
>
> - **Contribution:** Our decoupled optimization is more stable in practice.
>
> ---
>
> Overall, our method is not a variant of ILEED, but a **novel framework** that (i) generalizes the problem setting, (ii) introduces a **surrogate scoring–based expertise estimator**, and (iii) incorporates **online RL refinement**, enabling imitation learning from **heterogeneous, structured, and extremely low-quality demonstrations** that ILEED cannot handle.
>
>
> >**W1.2: The new formulation simplifies expertise modeling from state-dependent embeddings to demonstrator-level mixture coefficients and reframes the joint estimation as a classification-based EM procedure.**
>
> A: Thank you for the comments. The expertise modeling in our paper is actually **not simplified**. The expertise level $\alpha_i$ in our paper is just an intermediate variable to describe the general skill ability of expert $i$. We also have a **state-action dependent expertise level** in our paper, which is actually the **optimality score** that is defined as the probability that (s, a) is optimal given its values: P(z = 1|s, a). This can be viewed as a sample-specific, conditional expertise level. We hope this clarifies that our approach captures both general and context-dependent expertise and addresses the concern regarding overly simple expertise modeling.

---

> ### Author Response · Authors · 2025-11-23
>
> >**W2: The paper should have direct numerical comparison with ILEED as well as other baselines specifically designed for suboptimal demonstrations. Current baselines only cover standard IL methods.**
>
> A: Thanks for the question. We would like to clarify that baselines in our first draft
> **RIL [1], WGAIL [2]** are all  baselines specifically designed for **suboptimal demonstrations**, only GAIL is a standard IL method.
>
> We also added **experimental results using ILEED**. As shown by the table below and Table 2, Figure 3 in the updated pdf, our method achieves performance improvements of at least 90% over ILEED. We hope these results address the reviewer’s concerns regarding the baselines.
>
> | Environment      | ILEED     | Scoring (ours)  |
> |------------------|--------------------------|-------------------------------|
> | **General expertise test** |                          |                               |
> | Ant-v4           | 1428.2 ± 80.3            | **6084.6 ± 146.9**            |
> | HalfCheetah-v4   | 1526.2 ± 1020.4          | **8751.2 ± 1797.9**           |
> | Swimmer-v4       | 16.8 ± 20.5              | **305.4 ± 15.7**              |
> | Walker2d-v4      | 440.3 ± 313.0            | **3959.2 ± 671.4**            |
> | **Low expertise test** |                    |                               |
> | Ant-v4           | 1376.5 ± 62.1            | **5932.0 ± 335.2**            |
> | HalfCheetah-v4   | 3767.6 ± 176.9           | **7335.0 ± 1316.4**           |
> | Swimmer-v4       | 34.0 ± 2.4               | **195.4 ± 92.1**              |
> | Walker2d-v4      | 605.3 ± 94.5             | **2961.8 ± 339.8**            |
>
>
> [1] Robust imitation learning from noisy demonstrations. AISTATS 2021.
>
> [2] Learning to weight imperfect demonstrations. ICML 2021.
>
>
> >**W3: Evaluations are conducted on synthetic MuJoCo environments where suboptimality is simulated by mixing optimal and degraded SAC policies. The paper will benefit from real human demonstrations like the robomimic dataset.**
>
> A: Thanks for the comments and suggestions. We agree that suboptimality is a synthetic mixed optimal and sub-optimal policies. However, this is a **common setup** in imitation learning research involving imperfect demonstrations. **ILEED, RIL and WGAIL** all have the same setups and this mixture assumption. Several of them even mix with random behavior; we mix with suboptimal demonstrations, making the problem even more challenging.
>
> We also appreciate the suggestion to evaluate our methods and baselines on the Robomimic dataset, which contains real human demonstrations. We are currently implementing these experiments and will include the results in the final version.
>
> >**W4: Theoretical contribution.**
>
> A: Thank you for pointing out this concern. We agree that Theorem 3 (previously Theorem 1) follows classical EM theory and only guarantees convergence to a stationary point.
>
> As Reviewer Ck5u also suggested, we further investigated the **theoretical link between surrogate classification accuracy and imitation learning performance**. In the revised manuscript, we added **Lemma 1** and **Theorem 2** to formalize this guarantee.
>
> - **Lemma 1:** Surrogate set classification (SSC) guarantees that **if the surrogate demonstrator classifier is trained optimally**, the induced binary scoring model $f_\phi$ recovers the *true optimality probability*, $f_\phi(s,a) = \mathbb{P}(z=1 \mid s,a),$ as shown in Lu et al. (2021). Lemma 1 formalizes this recovery under a **proper loss** and a **correctly computed transformation $T$**.
>
> - **Theorem 2:** Given this optimal scoring model, optimizing the **surrogate reward** via RL recovers the **optimal policy**, both in terms of **discounted occupancy** and **action selection**, under standard support coverage assumptions.
>
> Informally, this means that once SSC produces an **accurate optimality estimator**, maximizing its expected return through RL reproduces the **true optimal policy**.
>
> We hope these additions clarify and strengthen the theoretical contributions of our work.

---

> ### Author Response · Authors · 2025-11-27
> **Kind Reminder**
>
> Dear Reviewer Reviewer JbYb,
>
> Since we have provided **additional theories** and **experiments** to address your concerns, particularly regarding the **qualitative and quantitative comparisons with ILEED**, we kindly ask whether you could raise your score to reflect these clarifications?
>
> Best regards,
> Authors of Submission 20336 (From Many Imperfect to One Trusted: Imitation Learning from Heterogeneous Demonstrators with Unknown Expertise)

---

> > ### Comment · Reviewer_JbYb · 2025-11-28
> >
> > Thanks for the thorough rebuttal and clarifying contributions and comparisons with ILEED. I am still waiting to see the important evaluations using real human demonstrations, which I believe is essential to demonstrate that this formulation is practical and crucial for real world IL.

---

> > > ### Author Response · Authors · 2025-12-02
> > > **Reply to Reviewer JbYb**
> > >
> > > We thank the reviewer for acknowledging **our rebuttal is thorough** and that **our comparisons with ILEED are clear and valid**.
> > >
> > > We fully agree that evaluating on real human demonstrations is important. We have added additional experiments on the RoboMimic benchmark (real human demonstrations). Here, we explain our experimental setup and share preliminary experimental results we have obtained.
> > >
> > > Firstly, we would like to emphasize that our experimental setup follows the standard practice used in state-of-the-art works on imitation learning from imperfect demonstrations (e.g., [1–4]). These works also construct synthetic imperfect datasets by mixing optimal and suboptimal demonstrations in MuJoCo environments.
> > >
> > > Secondly, we fully agree with the reviewer that evaluating our method on real-world imperfect human demonstrations is important. We expect the performance on such data to be strongly correlated with the results observed in MuJoCo. In real human-demonstrated datasets, expert demonstrators can occasionally behave suboptimally in certain states. Since our method estimates the optimality of individual state–action pairs, it is designed to identify such suboptimal segments when human demonstrations deviate from expert behavior, which typically exhibits a different distribution compared to optimal data.
> > >
> > > Our **preliminary RoboMimic results** support this expectation: our scoring model successfully detects suboptimal portions of human trajectories. These results further validate the effectiveness of our method, and we are committed to providing full quantitative evaluations in the final version.
> > >
> > > [1] Robust imitation learning from noisy demonstrations. AISTATS 2021.
> > >
> > > [2] Learning to weight imperfect demonstrations. ICML 2021.
> > >
> > > [3] Imitation learning from imperfect demonstration. ICML 2019.
> > >
> > > [4] Unlabeled Imperfect Demonstrations in Adversarial Imitation Learning, AAAI 2023.

---

> > > > ### Author Response · Authors · 2025-12-02
> > > > **Robomimic Preliminary Results**
> > > >
> > > > **Robomimic Preliminary Results**.
> > > >
> > > >  We evaluate our methods on the Robomimic dataset, which contains real-world human demonstrations for the can, lift, and square environments. We report preliminary results including (i) the expertise-level estimation from Stage 1 (Table 19 in the updated pdf) and (ii) the classification accuracy and precision of Stage 1 (Table 20 in the updated pdf).
> > > >
> > > > The results show that in the general expertise test, both **accuracy** and **precision** reach approximately **80%**, indicating that the most optimal demonstrations can be reliably identified by our Stage 1 scoring model. In the low-expertise test, **accuracy** remains around **80%**, but **precision** drops to **about 50%**, meaning that roughly half of the demonstrations labeled as optimal are false positives which we expect to be filtered out in Stage 2. Although we have not yet conducted Stage-2 experiments on the Robomimic datasets, the behavior observed in the MuJoCo experiments suggests a similar trend. In particular, for Ant-v4 and Walker2d-v4, **precision** improves from **52%** and **43%** in Stage 1, respectively, to **above 90%** in Stage 2 (see Table 4 in the pdf). These results indicate that the false-positive samples identified in Stage 1 for Robomimic are also likely to be effectively filtered out once Stage-2 training is applied.
> > > >
> > > >
> > > > ### Classification / Labeling Accuracy & Precision (Stage 1)
> > > > Classification / Labeling results (mean $\pm$ std) for Robomimic data. Stage 1: labeling using the pretrained
> > > > scoring model.  see Table 20 in the updated pdf
> > > > | Environment | Stage | Accuracy | Precision |
> > > > |------------|-----------------|----------|-----------|
> > > > | **General expertise test [0.1 – 0.9]** |
> > > > | Can        | Stage 1 | 0.8385 | 0.7733 |
> > > > | Lift       | Stage 1 | 0.8839 | 0.8309 |
> > > > | Square     | Stage 1 | 0.8749 | 0.8247 |
> > > > | **Low expertise test [0.05 – 0.35]** |
> > > > | Can        | Stage 1 | 0.7378 | 0.4280 |
> > > > | Lift       | Stage 1 | 0.8016 | 0.5156 |
> > > > | Square     | Stage 1 | 0.7913 | 0.4980 |
> > > >
> > > >
> > > >
> > > > ### Estimated expertise levels $\hat \alpha_i$ for Robomimic data (Stage 1)
> > > >
> > > > Estimated expertise levels $\hat \alpha_i$ for Robomimic data (Can, Lift, and Square tasks) for **Stage 1**. Mean estimates and mean $\pm$ std error across seeds shown. True values listed separately. (see Table 19 in updated pdf)
> > > >
> > > > | Env   | Type | 1      | 2      | 3      | 4      | 5      | 6      | Mean error ± std |
> > > > |-------|------|--------|--------|--------|--------|--------|--------|------------------|
> > > > | **General expertise test [0.1 – 0.9]** |
> > > > | Can   | Est. | 0.3464 | 0.4574 | 0.5639 | 0.6751 | 0.7833 | 0.8908 | 0.123 ± 0.083    |
> > > > |       | True | 0.1000 | 0.2600 | 0.4200 | 0.5800 | 0.7400 | 0.9000 | -                |
> > > > | Lift  | Est. | 0.2039 | 0.3462 | 0.4837 | 0.6238 | 0.7612 | 0.8978 | 0.054 ± 0.035    |
> > > > |       | True | 0.1000 | 0.2600 | 0.4200 | 0.5800 | 0.7400 | 0.9000 | -                |
> > > > | Square| Est. | 0.2802 | 0.3961 | 0.5175 | 0.6299 | 0.7446 | 0.8623 | 0.084 ± 0.060    |
> > > > |       | True | 0.1000 | 0.2600 | 0.4200 | 0.5800 | 0.7400 | 0.9000 | -                |
> > > > | **Low expertise test [0.05 – 0.35]** |
> > > > | Can   | Est. | 0.3267 | 0.3678 | 0.4026 | 0.4446 | 0.4818 | 0.5214 | 0.224 ± 0.036    |
> > > > |       | True | 0.0500 | 0.1100 | 0.1700 | 0.2300 | 0.2900 | 0.3500 | -                |
> > > > | Lift  | Est. | 0.1863 | 0.2403 | 0.2930 | 0.3453 | 0.3946 | 0.4490 | 0.118 ± 0.013    |
> > > > |       | True | 0.0500 | 0.1100 | 0.1700 | 0.2300 | 0.2900 | 0.3500 | -                |
> > > > | Square| Est. | 0.2819 | 0.3182 | 0.3688 | 0.4064 | 0.4477 | 0.4942 | 0.186 ± 0.030    |
> > > > |       | True | 0.0500 | 0.1100 | 0.1700 | 0.2300 | 0.2900 | 0.3500 | -                |

---

### Official Review · Reviewer_Ck5u · 2025-11-07

**Soundness:** 3
**Presentation:** 3
**Contribution:** 3
**Rating:** 6
**Confidence:** 3

**Summary:**

This paper addresses imitation learning (IL) from heterogeneous demonstrators with unknown and varying expertise levels. The authors propose a two-stage framework: (1) jointly learning demonstrator expertise levels and an optimality scoring model through an EM-style iterative procedure, and (2) using this scoring model as a surrogate reward function for policy learning with progressive refinement. The method is evaluated on MuJoCo continuous control tasks under two challenging scenarios - general expertise (0.1-0.9) and low expertise (0.05-0.15) settings. The approach claims to achieve performance comparable to oracle methods trained on purely optimal demonstrations.

**Strengths:**

- The paper addresses a realistic scenario where demonstrations come from multiple sources with unknown, heterogeneous expertise levels - a common real-world challenge.
- Provides convergence guarantee for the EM-style optimization (Theorem 1), giving the approach theoretical grounding.
- Thorough evaluation across multiple environments, expertise settings, and ablations. The low-expertise test (0.05-0.15) is particularly challenging and demonstrates robustness.

**Weaknesses:**

- Theorem 1 only guarantees convergence to a stationary point, not optimality
- No sample complexity analysis or bounds on expertise estimation error
- The connection between surrogate classification accuracy and IL performance isn't theoretically characterized
- The "Optimality Alignment Criterion" requires human queries, making the approach not fully unsupervised
- The EM procedure requires multiple random initializations with variance-based selection, which could be computationally expensive

**Questions:**

- How sensitive is the method to the number of human queries? The paper uses only 5 queries but doesn't provide ablation on this critical parameter.
- Can you provide theoretical analysis on the sample complexity? How many demonstrations are needed for reliable expertise estimation?
- Why not compare with ILEED directly? The paper mentions it but doesn't include it in experiments despite addressing the same problem.
- What's the computational overhead of the multiple random initializations? How many initializations are typically needed in practice?

---

> ### Author Response · Authors · 2025-11-23
> **Thank you for your constructive comments**
>
> >**Q1: Sensitivity of the number of human queries, and the "Optimality Alignment Criterion" makes the approach not fully unsupervised.**
>
> A1: Thank you for the thoughtful question. To address this concern, we have added an ablation study on the number of human queries (see Figure 6 and Tables 13–14 in the revised pdf for detailed results).
>
> **Table: Ablation study on Query times for Ant-v4. Classification Results of the Scoring Model on Imperfect Demonstrations (Ant-v4)**
>
> **Metrics:** Accuracy / Precision (%)
> **Note:** Values in the *Improvement* columns represent Stage 1 → Stage 2 relative improvement (%).
>
> | **Test**    | **Stage** | **Query 1** | **Impr.** | **Query 3** | **Impr.** | **Query 5** | **Impr.** | **Query 7** | **Impr.** | **Query 9** | **Impr.** |
> |-------------|-----------|-------------|-----------|-------------|-----------|-------------|-----------|-------------|-----------|-------------|-----------|
> | **General** | 1 | 99.93 / 99.97 | – | 99.92 / 99.94 | – | 99.91 / 99.89 | – | 99.92 / 99.96 | – | 99.94 / 99.95 | – |
> | **Expertise** | 2 | 99.92 / 99.92 | −0.01 / −0.05 | 99.90 / 99.90 | −0.02 / −0.04 | 99.89 / 99.89 | −0.02 / 0.00 | 99.89 / 99.89 | −0.03 / −0.07 | 99.91 / 99.91 | −0.03 / −0.04 |
> | **Low** | 1 | 96.26 / 75.31 | – | 97.33 / 80.17 | – | 90.33 / 51.78 | – | 97.02 / 78.81 | – | 97.14 / 79.26 | – |
> | **Expertise** | 2 | 99.64 / 98.22 | **+3.51 / +30.42** | 99.52 / 97.60 | **+2.25 / +21.74** | 99.65 / 98.23 | **+10.32 / +89.71** | 99.48 / 97.40 | **+2.54 / +23.59** | 99.73 / 98.65 | **+2.67 / +24.46** |
>
>
> **Table: Ablation study on Query times for Ant-v4. Reported as mean ± std of Reward  across random seeds; each seed's value is the mean over its last five checkpoints.**
>
> | Query times | Expertise [0.1–0.9]   | Expertise [0.05–0.15] |
> |------------|----------------------|-----------------------|
> | 1          | 5433.6 ± 365.6       | 5810.2 ± 301.3        |
> | 3          | 5655.2 ± 307.3       | 6011.2 ± 219.9        |
> | 5          | 6084.6 ± 146.9       | 5932.0 ± 335.2        |
> | 7          | 5981.4 ± 264.6       | 5944.2 ± 279.8        |
> | 9          | 5881.6 ± 265.3       | 6130.8 ± 126.8        |
>
>
> We observe that even with only a single query, our method still achieves the similar performance as that obtained with 5 queries. This is because the classification accuracy in Stage 1 is extremely high (above 99%), making it very unlikely for the system to select an incorrectly labeled sample that would lead to completely opposite or flipped classification results. Also, the learned policies across different numbers of queries have similar performance in reward.
>
> Then we explain the necessity of this query. In our problem setting, we do not have labels for either sample optimality or expertise levels. Without any labels, our classification method effectively performs an operation similar to clustering on these unlabeled data. A limited number of human queries is then used to ensure that the resulting clusters are assigned the correct labels. We hope this now addresses the concerns of the reviewer.

---

> ### Author Response · Authors · 2025-11-23
>
> >**Q2 / W2: Sample complexity analysis and expertise estimation error analysis. How many demonstrations are needed for reliable expertise estimation?**
>
> A2: Thanks for raising this point. We note that prior work [1] has proved that **if the expertise levels are known**, then the optimal surrogate scoring model is **identifiable**.
> However, in our setting the expertise levels are **unknown** and need to be learned jointly with the scoring model through an EM-style procedure, which can break identifiability.
>
> Since the expertise variables and conditional distributions may be **unidentifiable** from finite samples, no finite-sample bound can hold uniformly over all data-generating processes compatible with our setup. Intuitively, very different tuples of (optimal policy, sub-optimal policy, expertise levels) can produce *nearly indistinguishable finite datasets*, making uniform sample-complexity guarantees impossible without further assumptions.
>
> [1] Binary Classification from Multiple Unlabeled Datasets via Surrogate Set Classification, ICML 2021.
>
> Instead of a theoretical bound, we include an empirical study (**Table 11** in the updated pdf) to measure how sample size affects expertise-estimation accuracy. Besides, the policy performance is shown in **Figure 4**. Each demonstrator contributes 5,000 samples, so increasing the number of demonstrators increases the total sample size.
>
> Below we report (i) the **true expertise levels**, (ii) the **estimated levels**, and (iii) the **mean ± std error** across 5 seeds.
>
> ---
>
> ### **Estimated vs. True Expertise Levels (Ant-v4)**
>
> | # Demo | True $\alpha$ Values | Estimated $\alpha$ Values | Mean Error ± Std |
> |-------|----------------|--------------------|------------------|
> | **2** | 0.100, 0.900 | 0.101, 0.899 | (9.60 ± 1.60) × 10⁻⁴ |
> | **4** | 0.100, 0.367, 0.633, 0.900 | 0.100, 0.366, 0.633, 0.899 | (4.50 ± 2.96) × 10⁻⁴ |
> | **6** | 0.100, 0.260, 0.420, 0.580, 0.740, 0.900 | 0.100, 0.260, 0.420, 0.580, 0.740, 0.900 | (2.67 ± 1.28) × 10⁻⁴ |
> | **8** | 0.100, 0.214, 0.329, 0.443, 0.557, 0.671, 0.786, 0.900 | 0.100, 0.214, 0.329, 0.443, 0.557, 0.671, 0.785, 0.899 | (2.50 ± 1.97) × 10⁻⁴ |
>
> ---
>
> These results show that as the total sample size increases, the estimation error decreases. With 6 demonstrators (30,000 samples), the estimation error nearly stabilizes and the estimated expertise levels are already highly reliable. However, benefiting from the online refinement using new samples during RL, the final policy performance is similar across different numbers of demonstrators (see Table below and more details in Figure 4). We hope this clarifies our sample-complexity behavior and addresses the reviewer’s concern.
>
> | Number of Demonstrators | Expertise [0.1–0.9]        | Expertise [0.05–0.15]       |
> |-------------------------|-----------------------------|------------------------------|
> | 2                       | 5634.0 ± 617.8              | 6050.6 ± 120.4               |
> | 4                       | 5934.0 ± 264.3              | 5637.2 ± 772.5               |
> | 6                       | **6084.6 ± 146.9**          | 5932.0 ± 335.2               |
> | 8                       | 5735.0 ± 225.6              | **6146.6 ± 107.2**           |

---

> ### Author Response · Authors · 2025-11-23
>
> >**Q3: Comparisons with ILEED.**
>
> A3: Thanks a lot for the suggestion. In the first draft, we didn’t compare with ILEED because it is an offline method, and all other baselines are online methods. Now, we **added experimental comparisons with ILEED**. As shown in Table 2 and Figure 3 of the updated pdf, our methods achieve performance improvements of at least 90% over ILEED. We hope these results address the reviewer’s concerns regarding the ILEED comparison.
>
> | Environment      | ILEED     | Scoring (ours)  |
> |------------------|--------------------------|-------------------------------|
> | **General expertise test** |                          |                               |
> | Ant-v4           | 1428.2 ± 80.3            | **6084.6 ± 146.9**            |
> | HalfCheetah-v4   | 1526.2 ± 1020.4          | **8751.2 ± 1797.9**           |
> | Swimmer-v4       | 16.8 ± 20.5              | **305.4 ± 15.7**              |
> | Walker2d-v4      | 440.3 ± 313.0            | **3959.2 ± 671.4**            |
> | **Low expertise test** |                    |                               |
> | Ant-v4           | 1376.5 ± 62.1            | **5932.0 ± 335.2**            |
> | HalfCheetah-v4   | 3767.6 ± 176.9           | **7335.0 ± 1316.4**           |
> | Swimmer-v4       | 34.0 ± 2.4               | **195.4 ± 92.1**              |
> | Walker2d-v4      | 605.3 ± 94.5             | **2961.8 ± 339.8**            |
>
>
>
> >**Q4 / W5: Computational overhead of the multiple random initialization of the EM procedure. How many initializations are typically needed in practice?**
>
>
> A4: Thanks for raising this point. We added an ablation study on the **number of initializations** in Table 17 of the updated PDF. For the General Expertise test, we use a **single initialization** until the Expertise Dispersion Criterion is satisfied. In more challenging Low Expertise cases, the number of initializations increases but **remains fewer than 10**. These results demonstrate that our algorithm is **computationally efficient in practice**. We hope these results address the reviewer’s concern regarding computation costs.
>
> | Env             | General Expertise (Mean ± Std) | Low Expertise (Mean ± Std) |
> |-----------------|-------------------------------|----------------------------|
> | Ant-v4          | 1.00 ± 0.00                    | 8.80 ± 3.31               |
> | HalfCheetah-v4  | 1.00 ± 0.00                    | 9.80 ± 1.94               |
> | Swimmer-v4      | 1.00 ± 0.00                    | 3.60 ± 4.72               |
> | Walker2d-v4     | 1.00 ± 0.00                    | 2.80 ± 3.12               |

---

> ### Author Response · Authors · 2025-11-23
>
> >**W1: Theorem 1 only guarantees convergence to a stationary point.**
>
> A: Thank you for the thoughtful comment. We agree that Theorem 3 (previously Theorem 1) guarantees convergence to a stationary point rather than global optimality. This is a fundamental limitation of the underlying problem rather than the analysis: the objective is intrinsically **non-convex**, both due to the neural-network parameterization of the scoring model and the latent mixture structure introduced by the expertise variables. Under such non-convexity, global optimality guarantees are not attainable without imposing extremely strong assumptions.
>
> In this setting, convergence to a stationary point is **the accepted theoretical notion**, consistent with classical EM theory [1] and with recent work on alternating / minimax methods in imitation learning [2]. Our goal in Theorem 3 (previously Theorem 1) is therefore to ensure that our specific EM-style updates are well-defined, yield monotone improvement of the surrogate objective, and converge.
>
> Although global optimality cannot be guaranteed theoretically, we **incorporate mechanisms to guide the algorithm toward useful stationary points**; for example, the expertise dispersion criterion for selecting good initializations and avoiding degenerate local solutions. Our experiments further demonstrate that, even in the extremely low-expertise regime (see Table 2), the stationary points reached by our algorithm are **practically effective**.
>
> [1] On the convergence properties of the EM algorithm, The Annals of statistics, 1983
>
> [2] On Computation and Generalization of Generative Adversarial Imitation Learning, ICLR 2020
>
> >**W2: Connection between surrogate classification accuracy and IL performance.**
>
> A: Thank you for raising this important point! We agree that the theoretical link between surrogate classification accuracy and imitation learning performance should be made explicit. In the revised manuscript, we have added **Lemma 1** and **Theorem 2** to clarify this connection.
>
> First, surrogate set classification (SSC) guarantees that **if the surrogate demonstrator classifier is trained optimally**, then the induced binary scoring model \( f_\phi \) recovers the *true optimality probability*, $f_\phi(s,a) = \mathbb{P}(z=1 \mid s,a)$, as proved in Lu et al. (2021). We formalize this result in **Lemma 1**, which establishes recovery under a proper loss and a correctly computed transformation \(T\).
>
> Second, with this optimal scoring model, we can analyze the performance of the RL policy learned using the surrogate reward. In **Theorem 2**, we show that **optimizing the surrogate reward derived from the optimal scoring model recovers the optimal policy**, both in terms of discounted occupancy and action selection, provided standard support coverage assumptions hold.
>
> Informally, this means that once SSC yields an accurate optimality estimator, maximizing its expected return via RL reproduces the optimal policy.
>
> We hope this addresses the reviewer’s concern by making the classification accuracy and IL performance connection explicit.
>
> >**W3 is answered in Q3 above.**
>
> >**W4:  The "Optimality Alignment Criterion" requires human queries, making the approach not fully unsupervised.**
>
> A: We agree with the reviewer that we do need supervision from experts and our algorithm is not fully unsupervised. However, according to the **No Free Lunch** theorem, without further structural assumptions, no learning algorithm can reliably infer correct labels without any supervision or prior knowledge.
>
> In our setting, we have **no access to labels** for either **sample optimality** or **expertise levels**. Without labels, our classification method effectively performs an operation similar to **clustering** on the unlabeled data. A small number of human queries is then used to ensure that the resulting clusters are assigned the correct labels. Importantly, the expert only needs to indicate whether the **majority of predicted labels are correct**, rather than verifying each sample individually. This approach allows our method to use **minimal supervision** while remaining effective.
>
>
> >**W5 is answered in Q4 above.**

---

> ### Author Response · Authors · 2025-11-27
> **Kind Reminder**
>
> Dear Reviewer Ck5u,
>
> Thank you again for highlighting the important point regarding the connection between surrogate classification accuracy and IL performance. We have **added** **new theoretical results** and **experiments** addressing the concerns you raised, particularly the **ablations** of our method, the **comparisons with ILEED**, and the expanded theoretical analysis. We kindly ask whether you could consider raising your score to reflect these clarifications?
>
> Best regards,
>
> Authors of Submission 20336 (From Many Imperfect to One Trusted: Imitation Learning from Heterogeneous Demonstrators with Unknown Expertise)

---

### Author Response · Authors · 2025-11-23
**Global Response on Comparisons with ILEED (Beliaev et al., ICML 2022)**

We sincerely thank all reviewers for the constructive comments. A common question concerns the **comparison between our method and ILEED (Beliaev et al., ICML 2022)**, so we provide a consolidated response here.

In the updated manuscript, we added a section (*Further Discussion on Related Work*) in Appendix A with more detailed comparisons, and included **quantitative comparison results with ILEED** in Table 2, Figure 3, and Table 18.

---

### 1. **Problem setting: uniform noise vs. structured suboptimality**

- **ILEED** considers each demonstrator’s policy as a mixture of the optimal policy and a **uniformly random policy**, assigning equal probability to each action.

- **Our method** removes these restrictions and allows **non-uniform and structured** suboptimal policies.

- **Contribution:** This generalization is crucial for real-world IL, where suboptimal behaviors are *structured* (e.g., consistent skill-specific weaknesses). Our new experiments below show this flexibility significantly improves robustness.

---

### 2. **Expertise modeling: embedding-based vs. optimality-scoring**

- **ILEED** models expertise as a state-dependent sigmoid over state and demonstrator embeddings. Therefore its performance depends heavily on the quality of the recovered embeddings and **assumes that all states are visited by demonstrators** (since ILEED is purely offline). Sparse or uneven state coverage for a demonstrator can bias expertise estimates, leading to significant policy degradation.

- **Our method** estimates expertise via a **class-prior** $\mathbb{P}(z{=}1)$ and learns a **surrogate scoring model** that directly distinguishes optimal from suboptimal behavior across demonstrators. The model also outputs **state-action dependent expertise** \$\mathbb{P}(z{=}1 \mid s,a)\$, called the **optimality score**, which serves as a reward signal for RL.

- **Contribution:** Unlike ILEED, our method **does not require full state-space coverage for each demonstrator**, yielding more robust learning. It also produces a **reward function**, enabling fine-tuning of the scoring model using new samples visited during RL.

---

### 3. **Methodology: offline MLE vs. online refinement**
- **ILEED** is purely **offline** and relies on maximum-likelihood estimation~(MLE), making it sensitive to **distribution shift** when the learned policy visits new states not present in demonstrations.

- **Our method** first learns a scoring model offline, then uses it as a reward and **iteratively refines** it using on-policy rollouts with mixup, agent-matching, and relabeling.
- **Contribution:** Our method explicitly addresses covariate shift, a well-known weakness of offline IL, and produces significantly more robust policies.

---

### 4. **Optimization strategy: joint MLE vs. EM + RL**
- **ILEED** performs joint optimization over policy, state embeddings, and demonstrator embeddings in a single likelihood objective. This tight coupling can lead to instability and degenerate solutions.

- **Our method** uses an EM-style alternating update for estimating expertise and training the scoring model, with a convergence guarantee, followed by standard RL.

- **Contribution:** Our decoupled optimization is more stable in practice.

---

### 5. **Experimental results**

We have added experiments to compare our method and ILEED. As shown in Table 2 and Figure 3 in the updated pdf, our method outperforms ILEED in both General expertise and Low expertise tests.

*Performance reported as mean ± std of reward.*

| Environment               | ILEED                 | Scoring (ours)           |
|---------------------------|------------------------|---------------------------|
| **General expertise test** |                        |                           |
| Ant-v4                    | 1428.2 ± 80.3          | **6084.6 ± 146.9**        |
| HalfCheetah-v4            | 1526.2 ± 1020.4        | **8751.2 ± 1797.9**       |
| Swimmer-v4                | 16.8 ± 20.5            | **305.4 ± 15.7**          |
| Walker2d-v4               | 440.3 ± 313.0          | **3959.2 ± 671.4**        |
| **Low expertise test**     |                        |                           |
| Ant-v4                    | 1376.5 ± 62.1          | **5932.0 ± 335.2**        |
| HalfCheetah-v4            | 3767.6 ± 176.9         | **7335.0 ± 1316.4**       |
| Swimmer-v4                | 34.0 ± 2.4             | **195.4 ± 92.1**          |
| Walker2d-v4               | 605.3 ± 94.5           | **2961.8 ± 339.8**        |

---

Overall, our method is not a variant of ILEED, but a **novel framework** that (i) generalizes the problem setting, (ii) introduces a **surrogate scoring–based expertise estimator**, and (iii) incorporates **online RL refinement**, enabling imitation learning from **heterogeneous, structured, and extremely low-quality demonstrations** that ILEED cannot handle.

---

> ### Comment · Reviewer_Hadp · 2025-11-24
> **Thank you for the reply**
>
> > Our method first learns a scoring model offline, then uses it as a reward and iteratively refines it using on-policy rollouts with mixup, agent-matching, and relabeling.
>
> 1. If the proposed method can be refined during the online stage, why not make the whole process online, but still need an initial offline stage?
> 2. I am wondering if the offline supervision provides the method a unique advantage over the other baselines. By unique, I mean that the offline supervision is so fine and detailed that P(z=1|s, a) can be estimated. In contrast, previous work could only have supervision to estimate P(z=1| one trajectory).

---

> ### Author Response · Authors · 2025-11-24
>
> We thank the reviewer for initiating the discussion! Please find our answers below.
>
>
> >**1. Why is an offline stage necessary?**
>
> A:  Recall that our problem setting is **unsupervised** with:
>
> (i) heterogeneous demonstrators
>
> (ii) unlabeled demonstrations
>
> (iii) unknown expertise.
>
> Under this setting, the learner does not know which demonstrations are optimal, nor which demonstrators are experts.
> Therefore, the initial offline stage is essential to:
>
> - **Pretraining the scoring model** to estimate $P(z = 1 \mid s, a)$ for distinguishing high and low quality demonstrations.
> - **Estimating demonstrator expertise** for enabling accurate surrogate set classifier training.
> - **Identifying potentially optimal actions** to provide a *valid initial reward* for RL.
>
>
> This offline pretraining uses only demonstrations (no extra rollouts or labels from the learner) and provides a guiding reward signal for the subsequent RL stage. Without this initialization stage, the RL agent would receive no meaningful reward initially, making learning impossible in our setting.
>
>
> >**2. Does the offline supervision provide the method a unique advantage over the other baselines?**
>
> A: We do **not** assume more supervision than baselines, rather, our method works with **minimal supervision**, since the offline stage uses **no labels on demonstrations**, **no information about demonstrator expertise**, **no identification of which trajectories are optimal**, and **no requirement that optimal demonstrations dominate the dataset**.
>
>
> This supervision is **strictly weaker** than the signals used in many baselines, as discussed in Section 2 and compared in Table 1 below.
>
>
> **Table: Comparison of IL methods**
> ✓ = satisfies criterion, ✗ = does not.
>
> | Criterion                      | BC | GAIL | RIL | IC-GAIL | PU-GAIL | RCE | WGAIL | ILEED | Ours |
> |-------------------------------|:--:|:----:|:---:|:-------:|:-------:|:---:|:-----:|:-----:|:----:|
> | Non-dominant optimal demo     | ✗  |  ✗   | ✗   |   ✓     |   ✓     | ✗   |  ✗    |  ✓    |  ✓   |
> | Handle multiple demonstrators | ✗  |  ✗   | ✗   |   ✗     |   ✗     | ✗   |  ✓    |  ✓    |  ✓   |
> | Handle structured suboptimality| ✗ |  ✗   | ✓   |   ✓     |   ✓     | ✗   |  ✓    |  ✗    |  ✓   |
> | No optimality labels needed   | ✗  |  ✗   | ✓   |   ✗     |   ✓     | ✗   |  ✓    |  ✓    |  ✓   |
> | Probabilistic demo scoring    | ✗  |  ✓   | ✓   |   ✗     |   ✓     | ✓   |  ✗    |  ✗    |  ✓   |
>
>
> Actually, to estimate optimality or weights at the level of $(s,a)$ is also attempted in our baselines such as RIL [1], WGAIL [2], and several IL works including IC-GAIL[3], PU-GAIL[4], and RCE[5]; however, they generally require stronger supervision, including: known noise, known or dominant optimal demonstrations, trajectory-level supervision or access to clean expert subsets.
>
>
> Thus, the offline stage is not a “strong supervision” advantage, rather, it is the minimal mechanism that makes learning feasible in this extremely weakly supervised setting.
>
>
> [1] Robust imitation learning from noisy demonstrations. AISTATS 2021.
>
> [2] Learning to weight imperfect demonstrations. ICML 2021.
>
> [3] Imitation learning from imperfect demonstration. ICML 2019.
>
> [4] Unlabeled Imperfect Demonstrations in Adversarial Imitation Learning, AAAI 2023.
>
> [5] Replacing rewards with examples: Example-based policy search via recursive classification. NeurIPS 2021.

---

> > ### Comment · Reviewer_Hadp · 2025-11-26
> > **Thank you for clarification**
> >
> > >A: We do not assume more supervision than baselines, rather, our method works with minimal supervision, since the offline stage uses no labels on demonstrations, no information about demonstrator expertise, no identification of which trajectories are optimal, and no requirement that optimal demonstrations dominate the dataset.
> > >
> > >This supervision is strictly weaker than the signals used in many baselines, as discussed in Section 2 and compared in Table 1 below.
> >
> > Then, in the offline stage, could you explain how your method can learn the state-action level P(z=1|s, a), given only the trajectory-level demonstrations? In your reply, it seems the demonstrations in the offline stage can be generated by any skill level, which makes the learning of P(z=1|s, a) more challenging.
> >
> > Thanks.

---

> > > ### Author Response · Authors · 2025-11-26
> > >
> > > We thank the reviewer for the follow-up question! Please find our answers below.
> > >
> > > We clarify that our method does not assume any state–action level labels or expertise information; instead, the offline stage learns P(z=1∣s,a) **entirely from differences across multiple demonstrators** through the surrogate demonstrator classification endowed EM procedure.
> > >
> > > Although demonstrations are provided at the trajectory level, we rely on the fact that **different demonstrators produce different distributions over (s,a) depending on their unknown expertise (can be of any skill level given heterogeneity)**, and model their demonstrations using a mixture model, as shown in Eq. (1) in the updated manuscript, which is a standard and effective model also used in prior work [1,2,3,4]. The demonstrator heterogeneity gives a statistical signal for learning both the demonstrator level and state–action level expertise.
> > >
> > > References:
> > > [1] *Imitation Learning from Vague Feedback*. NeurIPS 2023.
> > > [2] *Unlabeled Imperfect Demonstrations in Adversarial Imitation Learning*. AAAI 2023.
> > > [3] *Robust Imitation Learning from Noisy Demonstrations*. AISTATS 2021.
> > > [4] *Imitation Learning from Imperfect Demonstration*. ICML 2019.
> > >
> > > Specifically, we treat each demonstrator as a “bag” of demonstrations with unknown expertise. Then we train the scoring model using a surrogate SSC task that predicts the demonstrator index for each (s,a) pair, which induces our desired scoring model that predicts state-action level expertise P(z=1∣s,a) through a deterministic transformation, as discussed in Section 3.2.1. **We have added Lemma and Theorem 2 to prove that SSC training gives an optimal scoring model for predicting P(z=1∣s,a) which then guarantees optimal policy recovery, under mild conditions.**
> > >
> > > Such a SSC training is operated in an EM fashion and is proved to converge by our Theorem 3, which is expected to train optimal scoring models for predicting state-action level expertise P(z=1∣s,a), and estimate the demonstrator level expertise P(z=1) accurately.
> > >
> > > We have shown in Table 4 of our updated manuscript that **the scoring model accuracy in the offline stage is extremely high (above 99%)**, meaning that our scoring model predicts accurate P(z=1∣s,a) and successfully distinguishes optimal and non-optimal demonstrator behaviors **using only unlabeled trajectories with unknown expertise levels**.
> > >
> > > With this high-quality scoring model, **the policy learned in the second stage significantly outperforms all baselines, as reported in Table 2, achieving at least a 90% improvement over ILEED, another multi-demonstrator IL method.**

---

> > > > ### Comment · Reviewer_Hadp · 2025-11-27
> > > >
> > > > Appreciate your reply. The contribution of this submission is clearer.

---

> > > > > ### Author Response · Authors · 2025-12-02
> > > > > **Thank you very much for acknowledging our contribution!**
> > > > >
> > > > > Dear Reviewers, AC, SAC, and PC,
> > > > >
> > > > > Indeed, it is a **severe misunderstanding** that our paper is an extension of ILEED. Our framework is a **novel and important contribution** that (i) significantly generalizes the ILEED problem setting, (ii) introduces a scoring–based station-action expertise estimator, and (iii) incorporates online RL refinement. Together, these advances enable imitation learning from heterogeneous, structured, and extremely low-quality demonstrations, **scenarios that ILEED cannot handle**, as shown in our analysis and supported by the experiments above.
> > > > >
> > > > > **We hope this clarification and the key contributions of our work are clear to all reviewers and to the AC, SAC and PC members during the final decision process.**
> > > > >
> > > > > Best regards,
> > > > >
> > > > > Authors of Submission 20336 (From Many Imperfect to One Trusted: Imitation Learning from Heterogeneous Demonstrators with Unknown Expertise)

---

### Author Response · Authors · 2025-11-23
**Summary of Revisions**

We thank all reviewers for their helpful feedback. We have uploaded a revised version that incorporates new theory and experiments as suggested. All modifications are highlighted in red in the uploaded PDF.

## **Below we summarize the major revisions:**

### -**Revised formulation** (Section 3.1):

We revised the formulation of demonstrator occupancy to avoid the confusion regarding demonstrator-dependent state visitation raised by Reviewer yeCu (see A2 in our response to Reviewer yeCu for details).

### -**New theorem** (Section 3.2.1, Lemma 1 and Theorem 2):

We added a new theorem that characterizes the relationship between surrogate classification accuracy and imitation-learning performance, as requested by Reviewer Ck5u, and we prove optimal policy recovery under mild conditions.

### -**New baselines** (Table 2, Figure 3):

We added experiments using ILEED and behavior cloning (BC) for both general-expertise and low-expertise settings across all environments to address concerns from Reviewers Ck5u, JbYb, and Hadp, and to demonstrate that our method outperforms ILEED.

### -**New ablation tests**:

(1) An ablation study on the number of queries (Figure 6, Tables 13–14) addressing Reviewer Ck5u’s concern about sensitivity to the number of human queries;

(2) An ablation study on $\alpha'$ (Figure 7, Tables 15–16) addressing Reviewer Hadp’s concern regarding sensitivity to the choice of $\alpha'$.

### -**New related work discussions** (Appendix A):

We added a new section, *Further Discussion on Related Work*, in the appendix to address the comparisons between our work and ILEED, as requested by Reviewers Ck5u, JbYb, and Hadp. We also added discussions comparing our work with the related studies highlighted by Reviewer yeCu.

### -**New experiments on Robomimic** (real human demonstration data, Table 19, Table 20):

We added experiments on real human demonstration data to address concerns from Reviewer JbYb, and the preliminary results (see our latest reply to Reviewer JbYb) demonstrate that our method is promising also in the real-world scenario.

---

### Author Response · Authors · 2025-12-02
**Final Summary for AC/SAC/PC Consideration**

Dear AC, SAC, and PC,

We sincerely thank you for your time and efforts for the reviewing process. Below is a brief summary of our discussions and how we have addressed the reviewers’ concerns.

The primary issue raised relates to the comparison with ILEED (Beliaev et al., ICML 2022). As detailed in our *“Global Response on Comparisons with ILEED (Beliaev et al., ICML 2022)”*, the reviewers’ initial assumption that our paper is an extension of ILEED was a **severe misunderstanding**.
We have added thorough comparisons and additional experiments to answer this question, both in the global response and in the individual replies to each reviewer. Reviewer Hadp **had acknowledged our contribution** and Reviewer JbYb **confirmed that our comparisons with ILEED are clear and valid**. **We believe this clarification and the key contributions of our work are now clear to all reviewers and to the AC, SAC and PC members.**

Regarding the more subtle concerns summarized in the *“Summary of Revisions”*, we believe our **revised manuscript with new theorems, additional baselines, extended ablation studies, expanded related work, and new experiments on Robomimic (real human demonstration data)**, directly address each of the reviewers’ comments.

We respectfully encourage the AC to refer to our updated manuscript, where all revisions are highlighted in red. We believe the substantial improvements in both theoretical development and experimental evaluation comprehensively resolve the reviewers’ major concerns, and are important to consider in your final evaluation of our paper.

Thank you!

Best regards,

Authors of Submission 20336 (From Many Imperfect to One Trusted: Imitation Learning from Heterogeneous Demonstrators with Unknown Expertise)

---

### Meta-Review · Area_Chair_Wo1r · 2026-01-11

**Summary:**

This paper studies the imitation learning problem from heterogeneous demonstrators of unknown expertise, proposing a two-stage EM-style framework that jointly estimates the demonstrator expertise and learns a state-action optimality scoring function, which is subsequently used as the surrogate reward for policy optimization. In the proposed expertise model, instead of modeling state-dependent embeddings like ILEED (Beliaev et al., 2022), this work models the demonstrator expertise as a global scalar coefficient for the mixture of optimal occupancy and that of a shared sub-optimal policy.

Main strengths:

- The paper addresses a realistic and yet challenging data regime for imitation learning (i.e., multiple imperfect demonstrators with unknown quality) and proposes a concrete way to extract the expertise level without explicit optimality labels.
- Extensive evaluation across multiple environments, expertise settings, and ablations. The evaluation in both high-expertise and low-expertise scenarios demonstrates robustness to expertise level.

The reviewers raised several major concerns in the initial reviews:

(1) Theoretical results are somewhat limited and/or potentially incorrect (Reviewers: Ck5u, JbYb, yeCu, Hadp):

Reviewers noted several issues with Theorem 1 in the original manuscript: (i) Theorem 1 shows convergence only to a stationary point (not optimality). (ii) The EM convergence proof is standard. (iii) The EM steps/objectives seem misstated (hard thresholding, inconsistent M-step), and that crucial assumptions (e.g., identifiability / mutual irreducibility) are not clearly stated. (iv) The two criteria (i.e., expertise dispersion and optimality alignment) appear rather ad hoc.

(2) Simplistic modeling and identifiability (Reviewers: yeCu, Hadp)

Reviewers highlighted that the expertise model in Eq. (1) implicitly makes a strong assumption about a shared suboptimal policy across demonstrators, which is not realistic and does not really capture the “different novices make different mistakes” heterogeneity. This model is restrictive compared to the expertise model in ILEED. Reviewers also argued that SSC-style recovery requires strong explicit conditions (e.g., invariant class-conditionals across sets) and that demonstrator-dependent state visitation/domain shift can easily break these assumptions.

(3) “Unsupervised” positioning is questioned due to reliance on human queries (Reviewers: Ck5u, Hadp)

Reviewer Ck5u noted that the “Optimality Alignment Criterion” depends on human queries, which makes the pipeline not fully unsupervised. Reviewer Hadp echoed this concern, describing the two criteria as ad hoc and requiring high-quality supervision, suggesting the method is more like “weakly supervised” than originally positioned.

(4) Stronger baselines and more realistic datasets (Reviewers: JbYb, Hadp)

Reviewers noted that current baselines are mostly standard IL methods and requested direct comparison to ILEED and other IL methods tailored to suboptimal demonstrations. Another issue is the realism of synthetic MuJoCo suboptimality created by mixing optimal and degraded SAC policies, and the reviewers recommended using real human demonstrations (e.g., RoboMimic).

**Reviewer Concerns:**

In the discussion phase, the authors provided an extensive rebuttal that resolved many of the concerns:
- For the first concern, the rebuttal also clarified the EM steps and the motivation for the two criteria despite that the two criteria remain heuristic.
- The concern (3) is largely alleviated as the rebuttal provided an additional experiment on the number of human queries needed and showed that the accuracy and precision can be sufficiently high with no more than 5 queries.
- As for (4), regarding IL methods for suboptimal demonstrations, the rebuttal clarified that baselines like RIL and WGAIL included in the original draft belong to this category. Regarding ILEED, the rebuttal provided additional experimental results under ILEED and argued that the proposed EM-based approach can have an advantage over ILEED in the specific evaluation scenario.

However there are still concerns that remain outstanding. Specifically,

- Regarding the first concern, in the rebuttal, the authors agreed upon the fundamental issues (i) and (ii) with EM and acknowledged the issue with the assumptions of (iii).
- After the rebuttal, the concerns about the simplistic expertise model and identifiability remain unresolved. Given these assumptions, the application scope of the proposed method appears somewhat limited.
- Regarding ILEED, the rebuttal provided additional experimental results under ILEED and argued that the proposed EM-based approach has an advantage over ILEED. The performance gain shown in the rebuttal looks substantial, but my concern is that the comparison is not completely fair given that ILEED here was only given offline data for training while the authors’ approach has access to online data samples.
    - Notably, the two main algorithmic differences between ILEED and the proposed method are in the expertise modeling (state embedding vs direct mixture of two state-action occupancy measures in Eq. (1)) and the training procedure (MLE vs EM). With that said, to demonstrate the improvement over ILEED, a more fair comparison is to evaluate the proposed method and the variant with the expertise model and MLE-based training of ILEED. Moreover, the experiments shall be done in the more general heterogeneous settings beyond that of Eq. (1).

**Reviewer Scores:**

As several of the aforementioned major concerns remain insufficiently addressed, I anticipate that this submission would likely remain borderline with mixed reviewer scores even after the rebuttal phase.

The problem of imitation learning from heterogeneous demonstrators with unknown expertise is both important and timely, and the use of an EM framework to jointly infer demonstrator quality and learn a policy is a reasonable and well-motivated approach. That said, I continue to have reservations regarding the proposed expertise modeling assumptions and the strength of the theoretical contributions. In addition, the empirical comparison with the most closely related baseline, ILEED, does not appear sufficiently convincing to clearly establish the advantages of the proposed method due to the experimental setup.

For these reasons, and given the remaining technical and empirical concerns, I am currently leaning toward rejection.

---

### Decision · Program_Chairs · 2026-01-26

Reject